# TEXT-TO-SQL BENCHMARKS FOR ENTERPRISE RE-ALITIES: UNDER MASSIVE SCOPES, COMPLEX SCHEMAS AND SCATTERED KNOWLEDGE

## ABSTRACT

Existing Text-to-SQL benchmarks remain overly idealized and differ substantially from enterprise scenarios, which require retrieving tables from massive query scopes, interpreting complex schemas, and locating scattered knowledge across large collections of documents. To address these gaps, we present two enterprise benchmarks, BIRD-Ent and Spider-Ent, constructed through a cost-effective refinement framework applied to their academic counterparts (BIRD and Spider), together with a new task paradigm, Dual-Retrieval-Augmented-Generation (DRAG) Text-to-SQL, which formalizes the dual-retrieval workflow of table schemas and knowledge documents prior to SQL generation. Our benchmarks exhibit three defining characteristics of enterprise settings: massive query scopes with over 4,000 columns, complex schemas with domain-specific and heavily abbreviated table and column names, and scattered knowledge distributed across enterprise-style documents totaling 1.5M tokens. These properties make the benchmarks substantially more realistic and challenging than existing ones. Evaluation on several state-of-the-art large language models (LLMs) reveals a sharp performance drop, with only 39.1 EX on BIRD-Ent and 60.5 EX on Spider-Ent, underscoring the gap between academic performance and enterprise requirements. By providing a rigorous and discriminative testbed under the DRAG Text-to-SQL paradigm, our benchmarks offer a valuable resource to advance research toward Text-to-SQL systems that are reliable and deployable in real-world enterprise environments.

## 1 INTRODUCTION

Converting natural language questions into SQL queries, commonly known as the Text-to-SQL task, aims to enable non-experts to interact with relational databases and assist professionals in writing SQL more efficiently. Owing to the widespread use of relational databases in real-world applications, Text-to-SQL has long been a topic of significant interest in both the NLP and database research communities (Li et al., 2023a; Pourreza & Rafiei, 2023; Gao et al., 2024; Shkapenyuk et al., 2025).

Benchmarks play a central role in advancing Text-to-SQL research, as they not only define the task setting but also serve as the foundation for measuring research progress and comparing approaches. Recently, LLMs have achieved impressive scores on academic benchmarks, for example, reaching an execution accuracy (EX) of 77.5 on BIRD (Li et al., 2023b) and as high as 91.2 on Spider (Yu et al., 2018). However, such results do not imply that the challenges of Text-to-SQL have been largely solved. As shown in Figure 1, most existing academic benchmarks, including Spider and BIRD, remain overly idealized, characterized by limited scope, simple schema, and inlined knowledge, all of which diverge significantly from the complexities of real-world enterprise practice. In response, several recent efforts have attempted to construct enterprise-oriented benchmarks, such as Spider 2.0 (Lei et al., 2025) and BEAVER (Chen et al., 2025). While these are important steps forward, they still fall short of capturing the realities of enterprise scenarios. Specifically, their *query scope* (i.e., the accessible schema scope for a single query) remains limited to dozens of tables, and the external knowledge is still provided as readily accessible snippets directly appended to queries, which is overly brief and sanitized compared to enterprise knowledge documentation.

Figure 1: Limitations of existing benchmarks in reflecting real-world enterprise challenges.

Based on our observations of real-world enterprise scenarios, enterprise Text-to-SQL presents challenges that go well beyond these benchmarks. First, the query scope is extremely massive in enterprise scenarios, typically represented by a data asset that contains hundreds or even tens of thousands of tables originating from diverse sources. Second, table and column names in enterprise schemas often carry domain-specific, complex meanings; they tend to be lengthy, heavily abbreviated, and difficult to align with the natural language used in user questions. Third, answering a question often requires external knowledge (e.g., column descriptions and domain knowledge), which is sparsely scattered across a large number of unstructured and heterogeneous documents. Instead of generating SQL directly from a question and a predefined schema, enterprise Text-to-SQL systems must first search over a broad query scope to retrieve the relevant tables, and search across large-scale document collections to locate useful knowledge. These realities underscore the urgency of constructing new benchmarks that can faithfully reflect the challenges faced in enterprise scenarios.

However, collecting data directly from real-world enterprise scenarios presents substantial challenges. Privacy concerns often prevent clients from sharing their data, and the cost of data curation, including cleaning, validation, and annotation, is typically prohibitive. To address these obstacles, we propose a **cost-effective benchmark refinement approach that builds upon existing academic benchmarks**. Our method progressively enhances the realism of these benchmarks along three dimensions. First, at the **domain level**, We expand the tables of the original databases by domain using LLMs, thereby shifting the query scope to massive and heterogeneous data assets. Second, at the **schema level**, we inject realistic enterprise noise into tables and columns. Finally, at the **knowledge level**, we convert query-specific knowledge snippets into large-scale external document collections, providing a more realistic external knowledge storage environment. Throughout the refinement process, we preserve the original benchmark's questions and SQL structures and the modifications are largely driven by LLMs with minimal human intervention, which significantly reduces the cost of constructing new benchmarks. To further align with enterprise Text-to-SQL workflows, we also introduce a new paradigm termed **Dual-Retrieval-Augmented-Generation Text-to-SQL**, which requires the model to retrieve information from large-scale data assets and external knowledge documents before generating the final SQL query.

Finally, by applying our refinement framework to two academic benchmarks, BIRD and Spider, and introducing the DRAG Text-to-SQL paradigm, we construct two enterprise benchmarks: **BIRD-Ent** and **Spider-Ent** (collectively referred to as the Ent-series benchmarks). BIRD-Ent expands the query scope to an average of **4,150.3** columns ($55.8\times$ larger than BIRD), and Spider-Ent to **4,053.3** columns ($164.7\times$ larger than Spider), highlighting the challenge of massive scopes. They further introduce more complex schema information and BIRD-Ent is paired with **1.5M** tokens of enterprise-style documents containing scattered knowledge. Evaluation results on several state-of-the-art (SOTA) LLMs reveal a significant performance drop on these refined benchmarks, indicating that current models struggle to handle the key challenges posed by enterprise Text-to-SQL scenarios. We summarize our contributions as follows:

- We introduce a new Text-to-SQL task setting, termed DRAG Text-to-SQL, that models the complex workflows encountered in real-world enterprise environments.

- We propose a comprehensive and fine-grained benchmark refinement framework that enables transforming existing academic benchmark into an enterprise benchmark at very low

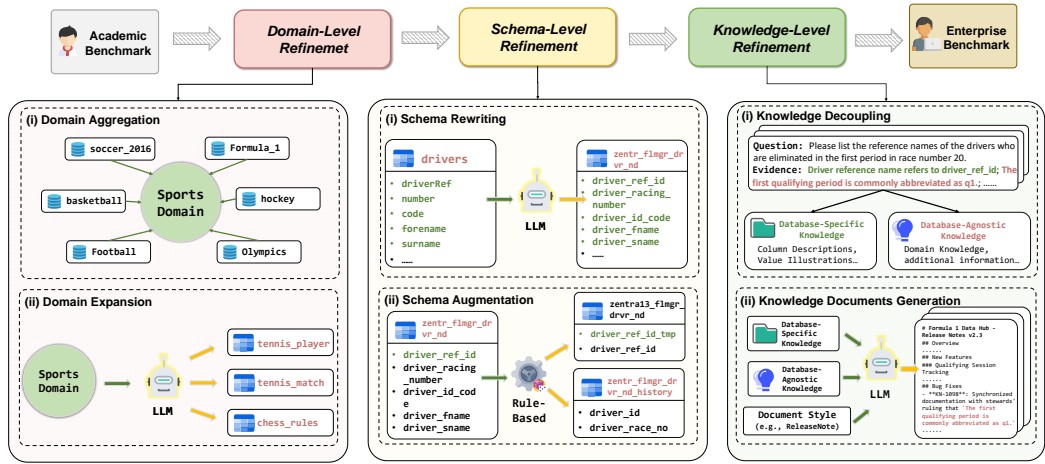

Figure 2: An overview of our benchmark refinement process.

cost. By applying this method to BIRD and Spider, we construct and release two high-quality enterprise Text-to-SQL benchmarks: **BIRD-Ent** and **Spider-Ent**.

- Our benchmarks reveal the formidable challenges of enterprise Text-to-SQL tasks, with SOTA LLMs achieving only **39.1%** EX on BIRD-Ent and **60.5%** EX on Spider-Ent.

## 2 TASK FORMULATION

Unlike previous benchmarks, where each question is paired with a predefined compact database and sometimes directly supplied with knowledge snippets, we define a new task paradigm, termed **Dual-Retrieval-Augmented-Generation (DRAG) Text-to-SQL**, which extends conventional Text-to-SQL by incorporating dual retrieval before SQL generation.

Given a natural language question $Q$, the system first retrieves the top-$k$ table schemas $\mathcal{C}_k$ from a large-scale data asset $\mathcal{D}$:

$$\mathcal{C}_k = \text{Retrieve}_{\text{schema}}(Q, \mathcal{D}), \tag{1}$$

and the top-$n$ external knowledge documents $\mathcal{K}_n$ from heterogeneous document collections $\mathcal{K}$:

$$\mathcal{K}_n = \text{Retrieve}_{\text{knowledge}}(Q, \mathcal{K}). \tag{2}$$

Finally, the target SQL query $Y$ is generated conditioned on the question and the retrieved context:

$$Y = f(Q, \mathcal{C}_k, \mathcal{K}_n \mid \theta). \tag{3}$$

Here, $\mathcal{C}_k$ denotes the retrieved table schemas, $\mathcal{K}_n$ denotes the retrieved knowledge documents, and $f(\cdot \mid \theta)$ denotes a model parameterized by $\theta$. This formulation explicitly reflects the dual-retrieval workflow inherent in real-world enterprise Text-to-SQL.

## 3 BENCHMARK REFINEMENT

As shown in Figure 2, our benchmark refinement consists of three key steps designed to simulate enterprise-level complexity: (i) **Domain-level refinement**, which enlarges the query scope by domain to reflect real-world data asset scales; (ii) **Schema-level refinement**, which introduces realistic naming and structural noise to mimic enterprise table schemas; and (iii) **Knowledge-level refinement**, which decouples query-specific external knowledge snippets from the input and embeds them across LLM-generated enterprise documents. We also incorporate quality control measures to ensure the reliability of the final benchmarks. All prompts used in this section, along with further details of the refinement methods, can be found in the Appendix B.

**Domain-level refinement.** Existing Text-to-SQL benchmarks restrict each query to a small, fixed database, making schema linking relatively simple. To simulate enterprise reality, we instead expand

the query scope by constructing domain-level data assets. We first group databases in academic benchmarks into topical domains and aggregate them into data assets. Each asset is then enlarged with LLM-generated tables designed in enterprise schema style, including realistic names, types, and constraints, while ensuring the question-SQL pairs in the original benchmarks remain valid.

**Schema-level refinement.** In enterprise settings, business-driven design and continual evolution yield long, structured, and often abbreviated names as well as pockets of redundancy. To capture this, we apply schema rewriting and augmentation. For rewriting, we convert original table names to hierarchical enterprise conventions (*<project>_<area>_<content>*) and abbreviate excessively long names using LLMs. We also rewrite columns by enriching them with clearer semantics while adopting enterprise-style abbreviations. For instance, *power* becomes *card_pwr_val*. For augmentation, we inject redundant tables and columns that resemble active ones to mirror real causes of redundancy in enterprises (e.g., legacy systems, migrations, testing). Such artifacts are often labeled with prefixes or suffixes like *history_* or *_migrated*, and they introduce ambiguity and stronger interference for table retrieval and schema linking.

**Knowledge-level refinement.** In enterprise scenarios, external knowledge must be retrieved from large document collections. To approximate this, we refine BIRD by first decoupling its query-specific evidence (knowledge snippet) from the queries (Spider does not incorporate external knowledge). The evidence contains both database-agnostic knowledge (e.g., domain knowledge) and database-specific knowledge (e.g., column descriptions). To avoid overlap with BIRD's database description files, we keep only the database-agnostic portion of the evidence and rely on the description files for database-specific content. Instead of presenting these materials directly, we segment them into passages and expand them with LLMs into longer documents written in enterprise styles (e.g., meeting minutes, technical manuals). Through an elaborate pipeline (more details in Appendix B.3.2). Finally, we construct 1,412 documents totaling about 1.5M tokens.

**Quality control.** Although each refinement step is carefully designed to minimize disruptions, error-free transformations cannot be guaranteed. To ensure benchmark quality, we verify three aspects: (i) answer uniqueness, checking whether a question admits only one semantically valid SQL regardless of syntactic form, as domain-level refinements may introduce semantically similar columns; (ii) semantic alignment, ensuring the question still matches its SQL after schema rewriting; and (iii) document correctness, confirming that generated knowledge documents faithfully preserve required content without semantic drift. We manually inspect at least 10% of samples, with double annotation and expert arbitration. Specially, for Spider, where many questions are underspecified, we observed frequent loss of answer uniqueness after refinement. We therefore conducted a manual full check, removed ambiguous cases, and retained 602 samples at last. Overall, BIRD-Ent reaches 94.8% answer uniqueness, 98.0% semantic alignment, and 96.7% document correctness, while Spider-Ent achieves 99.0% semantic alignment and 93.7% answer uniqueness, demonstrating that our benchmarks maintain high quality and that our refinement design effectively mitigates common errors.

## 4    DATA STATISTICS

We conduct an in-depth statistical analysis and comparison between existing Text-to-SQL benchmarks (including Spider, BIRD, BEAVER, and Spider 2.0) and our Ent-series benchmarks. As shown in Table 1, our benchmarks more closely reflect real-world enterprise Text-to-SQL scenarios across multiple critical dimensions.

**More massive query scopes.** Our Ent-series benchmarks substantially expand the query scopes compared to both academic (Spider, BIRD) and enterprise-oriented benchmarks (BEAVER, Spider 2.0). The average query scope in BIRD-Ent or Spider-Ent may span over 400 tables and 4000 columns, representing at least a 57.7× increase in table scope and a 55.8× increase in column scope over academic settings, and still at least 5.1× more tables and columns than existing enterprise benchmarks. This dramatic increase reflects the scale and complexity of real enterprise data environments.

**A more realistic knowledge storage environment and storage format.** BIRD-Ent benchmark explicitly model the decoupling between queries and external knowledge, simulating a real-world scenario where task-relevant information must be retrieved from a large pool of loosely organized documents. Unlike BIRD and Spider 2.0-snow/lite, which directly provide a document including a

| Benchmark | # Example | # Table / QS | # Col /QS | # Knowl. Tok. /Question | Enterprise Realism | Task Paradigm | Table Retrieval | Knowledge Retrieval |
|---|---|---|---|---|---|---|---|---|
| Spider | 2147 | 4.0 | 24.6 | - | ★ | Traditional | ✗ | ✗ |
| BIRD | 1789 | 6.8 | 72.6 | 25.0 | ★★ | Traditional | ✗ | ✗ |
| BEAVER | 203 | 77.5 | 708.4 | - | ★★★ | RAG | ✓ | ✗ |
| Spider 2.0-lite | 547 | 49.0 | 803.6 | 343.8 | ★★★ | Traditional | ✗ | ✗ |
| Spider 2.0-snow | 547 | 51.7 | 812.1 | 344.0 | ★★★ | Traditional | ✗ | ✗ |
| Spider 2.0-DBT | 68 | 21.4 | 337.7 | 1.3M | ★★★★ | Code Agent | - | - |
| BIRD-Ent | 1534 | 392.1 | **4150.3** | **1.5M** | ★★★★★ | **DRAG** | ✓ | ✓ |
| Spider-Ent | 602 | **413.0** | 4053.3 | - | ★★★★ | RAG | ✓ | ✗ |

Table 1: Comparison between existing benchmarks and our Ent-series benchmarks. QS denotes the Query Scope. Knowl. Tok. refers to the number of tokens of relevant knowledge.

small amount of dense external knowledge as part of the model input, BIRD-Ent adopts a sparsified storage format where relevant knowledge is scattered across longer, less structured documents, better reflecting enterprise realities. On average, BIRD-Ent requires the model to identify relevant information from over 1.5M tokens of candidate knowledge per question, approximately 4500× the size used in Spider 2.0-lite/snow, placing significantly greater demands on both retrieval accuracy and grounding robustness.

**A task paradigm more aligned with enterprise scenarios.** Unlike existing benchmarks that assume a predefined, compact database as input (e.g., BIRD, Spider 2.0-snow/lite), or BEAVER, which includes table retrieval but lacks knowledge retrieval, our Ent-series benchmarks adopt the DRAG Text-to-SQL paradigm (defined in Section 2) that decouples SQL generation from schema and knowledge selection. This mirrors the workflow of real-world enterprise analysts, who often operate over vast data and rely on heterogeneous knowledge sources to complete a task. By integrating both table and knowledge retrieval into the task paradigm, our benchmarks better reflect the multi-stage nature of such workflow, assess retrieval quality over large and noisy candidate spaces, evaluate model robustness under incomplete or imperfect inputs, and test whether generated SQL can be grounded in retrieved information-all of which are essential capabilities for deployment in enterprise scenarios.

## 5 EXPERIMENTS

### 5.1 IMPLEMENTATION DETAILS

During the benchmark refinement stage, we use Deepseek-R1-0528 (DeepSeek-AI, 2025) for synthesizing new tables within each domain, rewriting schemas and generating documents, due to its strong reasoning capabilities and excellent instruction-following performance. The process is conducted using the development sets and full database collections from BIRD and Spider, along with the database description files provided in BIRD.

During the evaluation stage, we adhere to the proposed DRAG Text-to-SQL paradigm. In the retrieval step, we adopt a standard embedding-based framework as a result of the ultra-long context caused by the massive query scopes and document collections. Specifically, each table schema and external knowledge document is encoded using a semantic embedding model and stored in a vector index. At inference time, the input question is embedded using the same model, and relevant candidates are retrieved via cosine similarity search in the vector space. The retrieved schemas, external knowledge documents, and the input question are then concatenated and fed into an LLM for zero-shot SQL generation. The prompt used in this stage is provided in Appendix.

### 5.2 EXPERIMENTAL SETUP

**Evaluation metrics.** In the evaluation setting of our Ent-series benchmark, the ability to retrieve the correct tables and external knowledge documents has a critical impact on the final SQL generation performance. Following standard practices in retrieval evaluation, we adopt precision, recall, and F1 score as our primary metrics. In addition, following BEAVER (Chen et al., 2025), we introduce the Perfect Recall (PR) metric, which measures the proportion of examples where all gold instances are included in the top-k retrieved candidates. This metric is particularly important, as failing to retrieve

| Model | EX | | | |
| --- | --- | --- | --- | --- |
| | BIRD | | Spider | |
| | dev | **Ent** | dev$^\dagger$ | **Ent** |
| GPT-4o | 57.6 | 34.4 (-23.2) | 84.4 | 56.8 (-27.6) |
| GPT-4o-mini | 53.6 | 26.4 (-27.2) | 83.4 | 53.8 (-29.6) |
| GPT-o4-mini | 58.3 | 39.0 (-19.3) | 83.9 | **60.5** (-23.4) |
| Qwen3-32B-no-thinking | 53.6 | 25.5 (-28.1) | 82.4 | 47.2 (-35.2) |
| Qwen3-32B | 59.2 | 33.8 (-25.4) | **84.7** | 57.8 (-26.9) |
| Qwen3-235B-A22B-no-thinking | 56.3 | 27.8 (-28.5) | 82.8 | 54.0 (-28.8) |
| Qwen3-235B-A22B | **59.6** | 36.9 (-22.7) | 82.4 | 58.5 (-23.9) |
| Deepseek-R1-0528 | 58.5 | **39.1** (-19.4) | 81.3 | 56.8 (-24.5) |

Table 2: The EX performance of various baselines on the Ent-series benchmarks as well as their original counterparts (the Spider dev set is **marked with** †, denoting the subset retained after the quality control described in Section 3, with 602 samples.). The model is provided with the top-10 tables and top-5 knowledge documents retrieved to obtain optimal results. The best results are highlighted in **bold**.

all relevant tables makes it highly unlikely for a model to generate the correct SQL. For the final SQL generation stage, we evaluate model performance using Execution Accuracy (EX), a widely adopted metric in prior work (Chang et al., 2023; Li et al., 2023b; Lei et al., 2025). EX measures the proportion of predicted SQL queries that yield the same execution results as their corresponding gold queries.

**Baseline models.** During the retrieval stage, we compare several SOTA embedding models, including Qwen3-Embedding-0.6B (Zhang et al., 2025), bge-m3 (Chen et al., 2024), and multilingual-e5-large-instruct (Wang et al., 2024). If not otherwise specified, we default to using Qwen3-Embedding-0.6B as the embedding model for the retrieval stage due to its superior retrieval performance. For the SQL generation stage, we select a range of leading LLMs, including the GPT series (4o, 4o-mini, o4-mini, OpenAI et al. (2024)), Qwen3-32B, Qwen3-235B-A22B (with and without thinking, Yang et al. (2025)), and Deepseek-R1-0528 (DeepSeek-AI, 2025). The selected models span both proprietary and open-source LLMs, as well as reasoning-augmented and standard variants, providing broad reference value.

## 5.3 MAIN RESULTS

We present our main evaluation results in Table 2, highlighting three core findings that offer deeper insights into model behavior under enterprise-level conditions.

**Existing LLMs exhibit unsatisfactory performance on enterprise Text-to-SQL tasks.** Moving from academic to enterprise settings, EX drops sharply on both benchmarks, with BIRD decreasing by up to 52.4% and Spider by up to 42.7%. Even the EX performance of strongest models on dev sets fall to 36.9 on BIRD-Ent and to 57.8 on Spider-Ent, indicating that today's high EX scores do not translate to enterprise-style conditions.

**Ent-series benchmarks faithfully capture the reasoning-intensive nature of enterprise scenarios.** Reasoning-enhanced variants such as Qwen3-32B and Qwen3-235B-A22B reduce the Ent-side drop by up to 5.8 points on BIRD and 8.3 points on Spider, while GPT-o4-mini shows the smallest declines overall (-19.3 on BIRD and -23.4 on Spider). In contrast, their non-reasoning counterparts degrade much more severely, underscoring that success on our benchmarks requires explicit reasoning to carefully handle massive query scopes, complex schemas, and scattered knowledge.

**Ent-series benchmarks increase separation among models.** On BIRD dev, the spread between the best and worst systems grows from 6.0 points on dev (59.6 vs 53.6) to 13.6 points on BIRD-Ent (39.1 vs 25.5). On Spider, the spread expands from 2.9 points on dev (84.2 vs 81.3) to 13.3 points

| Model | EX | | | | | | |
|---|---|---|---|---|---|---|---|
| | BIRD | | | | Spider | | |
| | dev | $\mathbf{Ent}_D$ | $\mathbf{Ent}_S$ | $\mathbf{Ent}_K$ | dev$^\dagger$ | $\mathbf{Ent}_D$ | $\mathbf{Ent}_S$ |
| Qwen3-32B-no-thinking | 53.6 | 44.2 (-9.4) | 41.7 (-11.9) | 42.4 (-11.2) | 82.4 | 72.8 (-9.6) | 64.3 (-18.1) |
| Qwen3-32B | 59.2 | 49.0 (-10.2) | 49.2 (-10.0) | 46.5 (-12.7) | 84.7 | 79.3 (-5.4) | 70.1 (-14.6) |
| Qwen3-235B-A22B-no-thinking | 56.3 | 46.7 (-9.6) | 45.2 (-11.1) | 42.6 (-13.7) | 82.8 | 75.6 (-7.2) | 64.3 (-18.5) |
| Qwen3-235B-A22B | 59.6 | 51.5 (-8.1) | 52.5 (-7.1) | 49.7 (-9.9) | 82.4 | 80.0 (-2.4) | 64.6 (-17.8) |
| Deepseek-R1-0528 | 58.5 | 51.0 (-7.5) | 56.5 (-2.0) | 49.9 (-8.6) | 81.3 | 78.6 (-2.7) | 65.6 (-15.7) |

Table 3: Ablation results on EX performance across different refinement stages. $\text{Ent}_D$, $\text{Ent}_S$, and $\text{Ent}_K$ represent benchmarks after domain-level, schema-level, and knowledge-level refinement, respectively. Top-10 tables and top-5 knowledge documents retrieved are provided to the model to obtain optimal results.

| Embedding Model | Table Retrieval | | | | | | | | Knowledge Retrieval | | | | | | | |
|---|---|---|---|---|---|---|---|---|---|---|---|---|---|---|---|---|
| | Top-5 | | | | Top-10 | | | | Top-5 | | | | Top-10 | | | |
| | P | R | F1 | PR | P | R | F1 | PR | P | R | F1 | PR | P | R | F1 | PR |
| BIRD-Ent | | | | | | | | | | | | | | | | |
| Qwen3-Embedding-0.6B | 28.7 | 76.8 | 40.8 | 62.5 | 16.4 | 86.0 | 27.0 | 76.5 | 29.6 | 46.2 | 34.4 | 14.9 | 20.8 | 62.9 | 30.2 | 31.8 |
| multilingual-e5-large-instruct | 26.0 | 70.1 | 37.0 | 52.8 | 15.1 | 79.8 | 24.9 | 66.4 | 28.7 | 43.9 | 33.1 | 12.8 | 21.1 | 62.2 | 30.4 | 31.8 |
| bge-m3 | 20.4 | 55.1 | 29.0 | 38.2 | 12.1 | 65.2 | 20.0 | 49.9 | 21.1 | 31.9 | 24.2 | 7.1 | 14.7 | 43.7 | 21.3 | 14.9 |
| Spider-Ent | | | | | | | | | | | | | | | | |
| Qwen3-Embedding-0.6B | 26.3 | 93.1 | 40.0 | 88.9 | 13.7 | 95.9 | 23.6 | 92.9 | - | - | - | - | - | - | - | - |
| multilingual-e5-large-instruct | 23.9 | 85.9 | 36.4 | 79.8 | 12.9 | 91.2 | 22.2 | 86.7 | - | - | - | - | - | - | - | - |
| bge-m3 | 18.3 | 65.8 | 28.0 | 57.1 | 10.4 | 74.3 | 18.0 | 66.2 | - | - | - | - | - | - | - | - |

Table 4: Retrieval performance for tables and external knowledge documents across the BIRD-Ent and Spider-Ent benchmarks using different embedding models. Since Spider does not include an external knowledge setting, we did not perform knowledge-level refinement, and therefore no knowledge retrieval results are reported.

on Spider-Ent (60.5 vs 47.2). This larger margin reveals capability differences that are obscured by traditional settings and makes the Ent-series benchmarks suitable for tracking real progress.

## 5.4 MORE ANALYSIS

**Each of the three refinement levels independently poses substantial enterprise-level obstacles for current LLMs.** Table 3 shows the ablation studies to compare the effects of different refinement stages. All three refinements lead to significant performance drops, confirming that each introduces non-trivial complexity. Among them, knowledge-level refinement ($\text{Ent}_K$) imposes the greatest challenge on BIRD, while schema-level refinement ($\text{Ent}_S$) has the largest impact on Spider. Consistent with our main results, models with explicit reasoning abilities demonstrate stronger robustness under refinement, and enterprise benchmarks amplify the performance gap between models, indicating a higher discriminative power in evaluating real-world readiness.

**Existing embedding-based retrieval frameworks are inadequate for the demands of enterprise-level Text-to-SQL retrieval.** We evaluate the performance of various embedding models in retrieving top-5 and top-10 table schemas and external knowledge documents. As shown in Table 4, retrieval precision remains consistently low across all models, particularly in the top-10 setting. Although recall is relatively high, the low precision indicates a significant amount of noise in the retrieved results. In particular, the low PR scores in table retrieval suggest that many downstream SQL generation tasks are grounded on incorrect or irrelevant inputs. Notably, all models perform worse on knowledge retrieval than on table retrieval, highlighting the greater difficulty of semanti-

cally grounding unstructured documents. These findings suggest that, despite their widespread use, current embedding-based retrieval methods fall short in supporting high-precision grounding under complex enterprise conditions, underscoring the need for more targeted retrieval techniques.

**Even under the oracle retrieval setting, current LLMs still struggle to handle the complexity and redundancy of enterprise-level schemas and knowledge documents.** We further examine the EX performance of different baselines on the Ent-series benchmarks under an oracle setting, where the input is restricted to only the gold table schemas and gold knowledge documents, with no distractors. The results are presented in Table 5. Despite the absence of retrieval noise, the EX scores remain unsatisfactory, especially on BIRD-Ent, where even the strongest model (Deepseek-R1-0528) reaches only 51.8. This indicates that performance bottlenecks persist even when correct

| Model | EX | |
|---|---|---|
| | BIRD-Ent | Spider-Ent |
| Qwen3-32B-no-thinking | 41.8 | 68.6 |
| Qwen3-32B | 46.9 | 70.0 |
| Qwen3-235B-A22B-no-thinking | 41.9 | 64.8 |
| Qwen3-235B-A22B | 50.7 | 64.6 |
| Deepseek-R1-0528 | 51.8 | 65.0 |

Table 5: EX performance under the oracle setting with gold-only table schemas and knowledge documents.

context is guaranteed, likely due to the inherent challenges in interpreting complex, redundant schemas and utilizing documents that contain abundant irrelevant information while the useful knowledge is highly scattered.

## 6 ERROR ANALYSIS

We randomly sample a total of 200 erroneous cases from BIRD-Ent and Spider-Ent for detailed error analysis, and categorize the common error types into 5 groups, the distribution of error types are shown in Figure 3.

**Schema errors (40.6%)** (i) Schema-retrieval errors (21.1%). Existing models frequently fails to retrieve the tables required to answer a question due to the massive query scopes and complex schemas in our benchmarks. (ii) Schema-linking errors (19.5%). Even if all of the gold tables are recalled, the model may still make mistakes in choosing the right tables and columns.

**Knowledge errors (30.8%)** (i) Knowledge-retrieval errors (24.8%). The knowledge documents required to answer the question may be absent from the retrieved ones. This is the most prevalent error type, indicating that the methodology used to retrieve knowledge documents requires improvement. (ii) Knowledge-grounding errors (6.0%). Even when the gold documents are retrieved, existing models may ignore or misinterpret the knowledge in the documents. This can be attribute to the fact that useful knowledge in our benchmark is often scattered across lengthy documents.

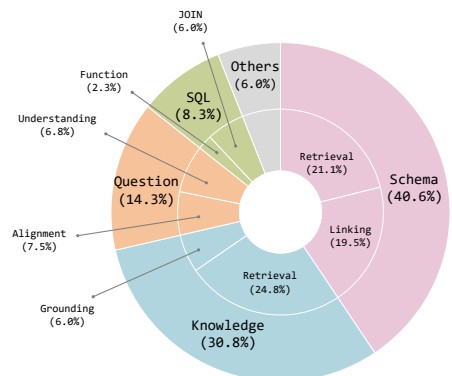

Figure 3: Error type distribution; case studies of each error type are provided in the Appendix C.

**Question errors (14.3%)** (i) Question-understanding errors (6.8%). In certain cases, the models will misinterpret the user's question and generate an irrelevant SQL. (ii) Question-alignment errors (7.5%). The SQL generated by the models fails to align with the user's requirements. For example, The user's question asks for *ID* and *name*, while the models only return the *ID*.

**SQL errors (8.3%)** (i) SQL-function errors (2.3%). The models occasionally make mistakes in the use of functions. For instance, when extracting substring, the models incorrectly use the SQLite-unsupported SUBSTRING(). (ii) SQL-JOIN errors (6.0%). In some cases, the model may join unnecessary tables or perform joins without following foreign key constraints.

**Other errors (6.0%)** Apart from the above cases, we recognize some other errors such as common-sense errors and value mismatch errors, which means that the literal chosen by the models do not match the value in the database. For example, the models may use *date* = '2007/01/02' to query the date, whereas the database stores the value of *date* in the format '2007-01-02'.

## 7 RELATED WORK

**Text-to-SQL benchmarks constructed from scratch** These benchmarks are typically released alongside the original datasets and databases. Among them, WikiSQL (Zhong et al., 2018) is the first large-scale cross-domain benchmark. Spider (Yu et al., 2018) expands the diversity and complexity of questions and SQLs under a cross-domain setting, and extends query scopes to multi-table databases. KaggleDBQA (Lee et al., 2021) incorporates database documentation into the parsing process, enriching the task formulation. BIRD (Li et al., 2023b) pushes benchmarks closer to real-world applications by introducing noisier database values and emphasizing the importance of external knowledge in query generation. Benchmarks such as SEDE (Hazoom et al., 2021), MIMIC-SQL (Wang et al., 2020), EHRSQL (Lee et al., 2022), and ScienceBenchmark (Zhang et al., 2023) address domain-specific challenges by constructing complex single-domain benchmarks grounded in practical use cases. Despite increasing in difficulty, these benchmarks remain relatively simple and idealized when compared to the capabilities and demands of modern LLM-based Text-to-SQL methods. As a result, recent efforts have shifted toward the construction of enterprise benchmarks. BEAVER (Chen et al., 2025) is the first benchmark specifically designed for enterprise Text-to-SQL scenarios. It constructs its databases and dataset based on real-world data warehouse environments, and introduces RAG task setting. Spider 2.0 (Lei et al., 2025), released later, focuses on even more complex database schemas and analytical tasks, incorporating multi-dialect challenges and agentic task formulations derived from enterprise use cases.

**Text-to-SQL benchmarks constructed by refinement** Due to the high cost of constructing benchmarks from scratch, many studies attempt to refine existing benchmarks by introducing special settings. Spider-DK (Gan et al., 2021b) enhances the knowledge dimension by defining and integrating five types of domain knowledge into the Spider development set, aiming to assess the generalization ability of Text-to-SQL models. Spider-Realistic (Deng et al., 2021) and Spider-Syn (Gan et al., 2021a) introduce noise by replacing explicit schema-related terms in natural language questions with synonyms. ADVETA (Pi et al., 2022) proposes Adversarial Table Perturbation (ATP), which focuses on evaluating model robustness under table-side disturbances by replacing column names with synonyms and adding confusing columns. Dr. Spider (Chang et al., 2023) further introduces 17 types of perturbations covering databases, natural language questions, and SQL queries, providing a comprehensive robustness evaluation. SParC (Yu et al., 2019) brings in a multi-turn interaction setting, guiding the construction of thematically consistent follow-up questions based on those from Spider. Spider-SS&CG (Gan et al., 2022) decomposes questions and SQLs in Spider into clauses and recombines them to construct a benchmark focused on clause-level compositional generalization, testing how well models generalize to new combinations of components seen during training.

Our Ent-series benchmarks automatically refine existing benchmarks around three key challenges inherent to enterprise Text-to-SQL tasks. By minimizing annotation costs, BIRD-Ent and Spider-Ent deliver challenging benchmarks that faithfully reflects real-world enterprise scenarios.

## 8 CONCLUSION

We release BIRD-Ent and Spider-Ent, two enterprise Text-to-SQL benchmarks that feature massive query scopes exceeding 4,000 columns, complex schemas, and scattered knowledge across documents with 1.5M tokens. Alongside these datasets, we introduce the DRAG Text-to-SQL paradigm, refelecting the real-world enterprise workflow. Together, they mirror the challenges faced in enterprise and expose substantial performance gaps in SOTA LLMs. The Ent-series benchmarks under the DRAG paradigm provide a rigorous and discriminative testbed for evaluating robustness, retrieval, and grounding abilities, offering a valuable resource for advancing Text-to-SQL research toward models that are reliable and deployable in real-world enterprise scenarios.

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

# A EXAMPLES

## A.1 REAL-WORLD DATA ASSET

Figure 4 shows an example of data asset schema from real enterprise scenarios. Sensitive information was anonymized during the review process. In real scenarios, the column and table descriptions provided in this example are often missing; we include them here solely for ease of understanding.

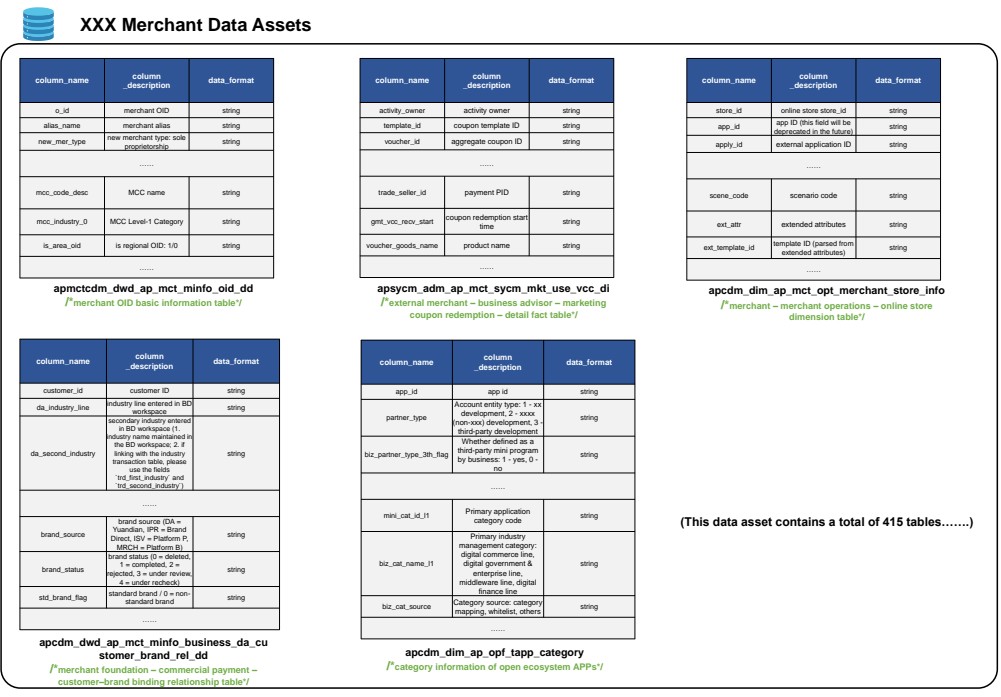

Figure 4: An example of enterprise data asset.

## A.2 ENT-SERIES BENCHMARKS

### A.2.1 DATA ASSET

Figure 5 shows an example of the data asset in our BIRD-Ent benchmark. The data assets constructed in our benchmarks simulate real enterprise assets in both naming conventions and scale. It is worth noting that real enterprise data assets' tables (see in Figure 4) often include partitioning information (e.g., *dd*) and data warehouse layering information (e.g., *dwd*). In our schema rewriting step, however, such details are omitted, as the refined benchmarks are built on SQLite databases and we are concerned that forcibly introducing them would cause confusion for benchmark users.

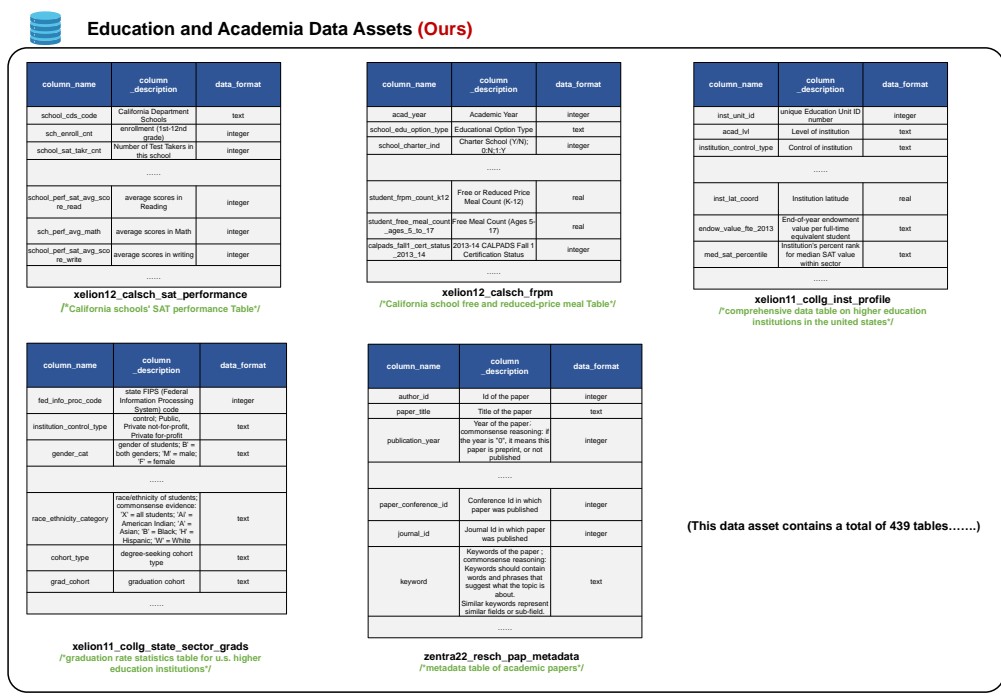

Figure 5: An example of BIRD-Ent's data asset.

### A.2.2 KNOWLEDGE DOCUMENT

Figure 6 shows an example of an external knowledge document in our BIRD-Ent benchmark. The document is presented in an FAQ style and contains column descriptions and value illustrations for a subset of the tables. Some content is omitted in the example due to the document's length.

## B DETAILS ON BENCHMARK REFINEMENT

### B.1 DOMAIN-LEVEL REFINEMENT

To simulate the way data tables are typically stored in enterprises, we introduce a domain-level refinement. In real-world enterprise settings, data tables are usually organized by domain to facilitate topic-specific analysis by data analysts. To mimic this practice, we first leverage LLMs to classify and aggregate the original databases in BIRD and Spider by domain, thereby constructing domain-specific data assets. The resulting classification of BIRD is shown in Table 6. Since our refinement is built upon the development sets of BIRD and Spider, we omit domains that are not covered in their development sets.

After aggregation by domain, the query scope is expanded, but it still falls far short of the enterprise scale, where a single data asset may comprise hundreds of tables. To bridge this gap, we perform domain expansion, prompting the LLM to generate new databases within a given domain. The

---

**Geographical Organization Data Management FAQ**

# 🌐 Geographical Organization Data Management FAQ

> **Introduction**
This document addresses common questions about managing geographical organization data in our global reference system. It covers data definitions, usage scenarios, and troubleshooting for tables tracking international organizations and their memberships.
◇ *New team members should review this before interacting with the `zentra25_georef` dataset.*

---

## 🏷️ Organization Metadata

### **Q1: How do we uniquely identify organizations in our system?**

We use two complementary identifiers:
- `organization_abbrev`: Stores the organization's abbreviation (e.g., **ASEAN** for the Association of Southeast Asian Nations).
- `city_name`: Records the **full name** of the organization.

☑ Always cross-reference both fields when verifying organizational identity.

---

### **Q2: Where is headquarters location data stored?**

Headquarters details are captured through three fields:
- `City`: The city where the headquarters are located.
- `province`: The province/state/region of the headquarters.
- `country_code`: The country code (ISO 3166) where the headquarters are located.

⚠ **Important**: Always validate province–country combinations to avoid mismatches like *"California, Canada"*.

……(Some content of the document is omitted here)……

🛠 Steps to resolve:
1. Verify spelling of `Organization` against `organization_abbrev`
2. Check `ORG_HISTORICAL_ALIAS` for legacy codes
3. If unresolved, run script: `OrgIntegrityCheck.py`

---

### **Scenario 2: Discrepancy between `city_name` and physical headquarters location**

🔁 Remember:
- `city_name` = Full **organization name**, NOT a geographic location.
- Physical HQs are only in the `City` field.

▷ Action:
- Flag any entry where `city_name` includes place names (e.g., `"Nairobi Agreement Council"`) for correction.

---

### **Scenario 3: Missing Type classifications in membership records**

🗋 Since `Type` lacks formal definition:
- Do **not** infer meaning from blank or populated values
- Tag affected records with `TYPE_UNDEFINED` in audit logs
- Escalate per **Section 4.3 of the Data Governance Policy**

---

## 🗒 Appendix: Change History

| Date    | Change Description                           | Ticket/Ref    |
|---------|----------------------------------------------|---------------|
| 2023-11 | Deprecated `Region` field                    | GLB-223       |
| 2022-08 | Rejected `Membership_Fee` field proposal     | RFC-771       |
| 2020-05 | Province naming standardization              | DATA-441      |
| 2018-02 | Country code standardization (FIPS → ISO 3166) | COMPLIANCE-98 |

---

> ⓘ This document is maintained by the Global Reference Data Team.
> Last updated: Based on changes up to **2023-11**.

Figure 6: An example of BIRD-Ent's external knowledge document.

tables from these newly generated databases are then added to the corresponding domain asset. We generate entire databases before extracting their tables, rather than generating tables directly, in order to ensure inherent logical connections among the tables, reflecting enterprise reality, where some tables within a data asset may originate from the same project or database. Moreover, in

| Benchmark | Domain | Databases |
|---|---|---|
| BIRD | retail_and_e-commerce | car_retails, retails, retail_world, regional_sales, retail_complains, sales, sales_in_weather, super-store, debit_card_specializing |
| | finance_and_economy | coinmarketcap, financial |
| | education_and_academia | authors, books, book_publishing_company, citeseer, college_completion, university, computer_student, cs_semester, language_corpus, student_loan, california_schools, student_club |
| | data_science_and_technology | codebase_comments, image_and_language, talkingdata, codebase_community |
| | healthcare_and_bioinformatics | synthea, genes, mental_health_survey, thrombosis_prediction, toxicology |
| | sports_and_athletes | hockey, ice_hockey_draft, european_football_1, soccer_2016, olympics, professional_basketball, european_football_2, formula_1 |
| | entertainment_and_media | disney, movie, movies_4, movie_3, movielens, movie_platform, simpson_episodes, law_episode, shakespeare, video_games, superhero, card_games |

Table 6: Domain aggregation results of the BIRD databases.

designing the prompts, we provide the model with the databases already present in the domain asset to prevent semantic collisions, which could otherwise lead to non-unique answers in the original datasets and degrade the quality of the final benchmarks. We also include examples of real enterprise table schemas in the prompt to guide the model in mimicking authentic schema styles. An example prompt for the domain expansion step is shown in Figure 14.

It is worth noting that we do not generate any sample values for the synthetic tables, nor do we verify their logical consistency or construct corresponding databases. This is because the original question-SQL pairs do not reference these tables, and their execution results are unaffected by them. The schemas of the synthetic tables serve as pseudo-schemas, they are designed solely to simulate enterprise scenarios by expanding the domain and introducing challenge to the schema linking process.

## B.2 SCHEMA-LEVEL REFINEMENT

At the schema level, the complexity of academic benchmarks is generally not enough compared to real-world enterprise databases. The enterprise-level database fields are often more intricate and widely abbreviated. In addition, in enterprise systems, the processes of maintenance and iterative updates frequently generate historical, temporary, or backup tables and columns. Although they are no longer actively used, they remain within the system and often exhibit strong similarities to certain existing tables. We collectively call them redundant tables and columns.

To better simulate these real-world challenges, we propose schema rewriting and schema augmentation strategies for academic benchmarks at the schema level. The first step is schema rewriting. For table names, enterprise scenarios often have naming conventions. Motivated by common enterprise practices, we propose a hierarchical naming convention *<project>_<area>_<content>* to simulate the enterprise environment. In this convention, *<project>* refers to the id of the project in enterprise scenarios; *<area>* is a generalization of the business area to which a table belongs; *<content>* is a summary of the content of the table. To transform the original table names into the desired format, we first provide LLM with available database information to generate the project and area name. Then, the model produces an abbreviated representation of the original table name and concatenate it with the previously generated project and domain names to construct a fully structured table name. The prompt of table name rewriting is shown in Figure 15 and Figure 16. For column

names, considering the high degree of abbreviation observed in enterprise schema, we propose a CoT process to rewrite column names by using LLM. First, we generate a concise, domain-aware contextual summary for each database using available schema information and database descriptions (when present). Its prompt can be seen in Figure 17. Second, by using the domain context together with the columns' descriptions within its table schema, we expand the column names. Its prompt can be seen in Figure 18. Last, to emulate enterprise terseness, we will abbreviate the column names with LLM. Details prompt is illustrated in Figure 19. For example, in the *card game* database, the table name *card* is rewritted as *zentra11_mcard_crd_catalog*, where *zentra11* is the project id of the enterprise, *mcard* represents the table belongs the card game area, and *crd_catalog* indicates that the table contains card catalog information, and the column name *power* (representing the power value of a card) is rewritten as *card_pwr_val*, a longer, more specific, and more heavily abbreviated field name.

The injection of redundant tables and columns is achieved in the following way: We randomly select a subset of tables from each database and, following the table-naming convention, generate redundant table names by appending common noise suffixes or prefixes. For each selected table, we clone its schema to create the corresponding noise table. Then we randomly add some columns related to the table and remove a few columns that were not primary keys or foreign keys from this table to simulate the subtle differences between the noise table and the existing table caused by database evolution. Finally, we append the noise columns, which are created by copying the existing column and adding common noise suffixes or prefixes to the original table. We control the ratio of redundant tables to original tables to be 1:4 to simulate the real environment of the enterprise.

| | **Table** | **Column** |
|---|---|---|
| **Suffix or Prefix** | bak, hist, drop, tmp, mid, snap-shot, history | backup, tmp, migrated,legacy, deprecated |

Table 7: Common Suffixes and Prefixes Summary.

## B.3 KNOWLEDGE-LEVEL REFINEMENT

BIRD acknowledges the necessity of external knowledge in Text-to-SQL tasks, since user queries are often concise and may naturally omit information that is crucial for answering the question (e.g., domain-specific knowledge or descriptions of database contents). However, in BIRD, such knowledge is directly appended as part of the query, which is unrealistic in enterprise scenarios where relevant information must instead be retrieved from large-scale external document collections. Moreover, much of the external knowledge in BIRD is overly simplistic, which rarely appears in enterprise environments. To address these issues, our knowledge-level refinement consists of two steps: knowledge cleaning and decoupling, and document generation.

Figure 7: An example of a BIRD database description file (for the *Paper* table in the *authors* database)

### B.3.1 KNOWLEDGE CLEANING AND DECOUPLING

In the cleaning step, we manually examined the external knowledge provided in BIRD and categorized its types (as illustrated in Figure 8). This included redundant knowledge such as common-sense facts, basic arithmetic, fundamental SQL syntax, trivial synonyms, and simple reasoning, all of which were identified and removed.

For the remaining non-redundant knowledge, our analysis revealed two main categories: database-specific and database-agnostic. Database-specific knowledge typically consists of column descriptions and value illustrations, most of which originate from the database description documents released alongside BIRD datasets. As exemplified in Figure 7, these documents cover nearly all

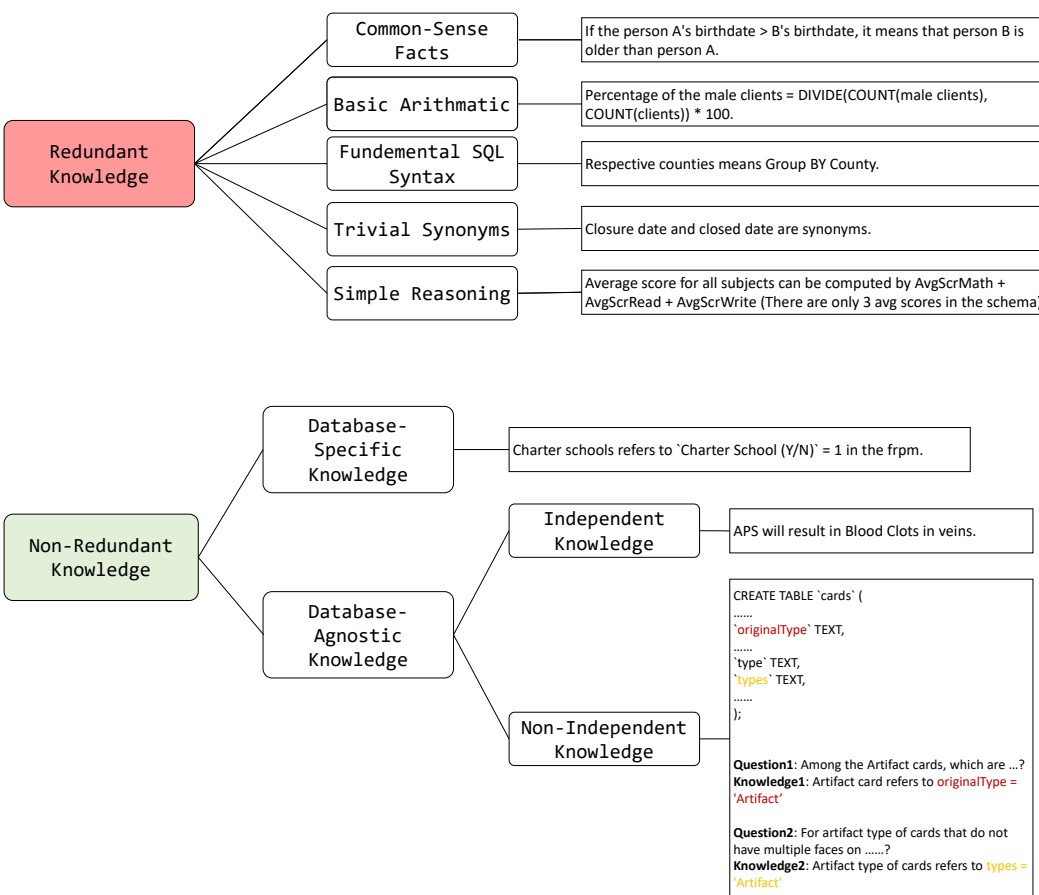

Figure 8: Classification and examples of BIRD external knowledge

columns across BIRD's databases, including their column descriptions, types, and value illustrations. We merged this category of knowledge with the corresponding database descriptions, treating them as source material for generating external knowledge documents, which more closely reflects enterprise settings where such information is typically scattered across heterogeneous documents.

Database-agnostic knowledge can be further divided into query-dependent and query-independent. Query-dependent knowledge supplements a specific question but is invalid in isolation. For example, as shown in Figure 8, the same noun (*Artifact cards*) may refer to different columns (*types* =' Artifact' or *originalType* = 'Artifact') across queries; such knowledge only makes sense when tied to the query context. Therefore, we attach it directly to the original query rather than treating it as external knowledge. In contrast, query-independent knowledge remains valid outside the original query context. Owing to its generality, this category is retained as additional source material for generating external knowledge documents.

### B.3.2 DOCUMENT GENERATION

The pipeline for generating knowledge documents is illustrated in Figure 9. In the first step, we segment the database description documents into chunks, each containing several columns along with their names, descriptions, types, value illustrations, and sampled values. These chunks are then combined with the independent external knowledge associated with the corresponding columns to form the raw external knowledge.

In the second step, we randomly select one of the predefined enterprise document genres (see in Table 8) and prompt DeepSeek-R1-0528 to generate a document in the chosen genre, where the raw external knowledge is naturally and contextually embedded. To facilitate subsequent inspection and

correction, we further instruct the LLM to annotate the generated text with special tags for each piece of knowledge content: $< ocn_i >$ for column names, $< cd_i >$ for column descriptions, $< vd_i >$ for value illustrations, and $< ek_i >$ for independent external knowledge. These annotations align the knowledge with standardized tags, enabling us to locate and verify the correctness of the embedded content. The prompt for this step is shown in Figure 20.

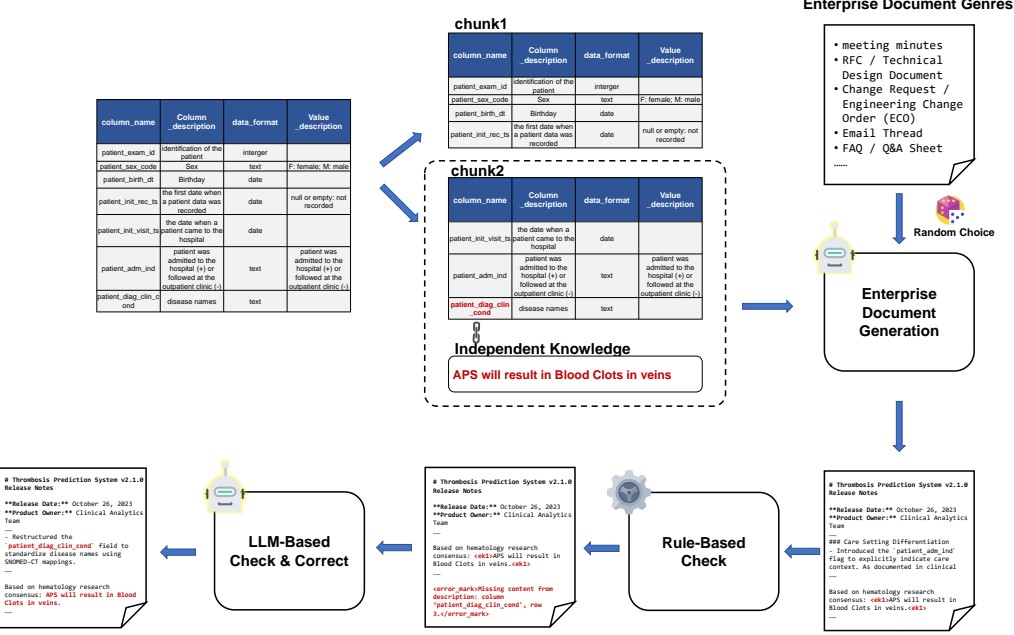

Figure 9: The pipeline for generating enterprise documents from raw external knowledge

In the third step, we perform rule-based checks on the generated documents to ensure: (i) all required tags are present, (ii) tags follow the correct format, and (iii) no unintended tags are included. If violations are detected, an $< error\_mark >$ tag is appended to the corresponding content, preparing the documents for LLM-based self-correction.

In the fourth step, we provide the annotated knowledge documents, the raw external knowledge, and the document generation rules as inputs to the LLM, which is tasked with correcting the documents. The prompt used for this step is illustrated in Figure 21. In the quality control step, we also leverage these pre-generated tags when inspecting the knowledge documents.

Through this elaborate pipeline, we accurately transform BIRD's external knowledge into a large collection of enterprise-style knowledge documents.

## C  CASE STUDY

Figure 10 illustrates a typical schema-retrieval error. The failure of the predicted SQL stems from the fact that not all target tables were included in the retrieved schema. When the model is not provided with the complete set of relevant tables, it becomes exceedingly difficult to generate a correct SQL query.

Figure 11 reflects another type of schema error. In this case, although the target table is all retrieved in the previous stage, the model does not select all the right tables and columns. This situation may be caused by the semantic similarity among tables and columns within the same domain and the interference of redundant tables. In the error analysis, such cases are not uncommon, suggesting that existing models may still exhibit limitations in their sensitivity to distinguishing among different tables and columns.

| Genre | Description |
|---|---|
| meeting minutes | Meeting minutes are written records of a meeting, typically including the time, location, attendees, topics discussed, decisions made, and action items for follow-up. It serves as an efficient communication tool to ensure that meeting participants and those who missed the meeting can stay informed and track the execution of decisions. |
| RFC / Technical Design Document | A Request for Comments (RFC) or Technical Design Document is used to propose and discuss technical decisions, design solutions, and system specifications. It provides a detailed explanation of the problem, proposed solutions, alternatives considered, risks, and potential impacts. This document serves as a reference to guide teams through the decision-making process, ensuring alignment and consistency in implementation. |
| Change Request / Engineering Change Order (ECO) | A Change Request (CR) or Engineering Change Order (ECO) is a formal document used to propose and approve changes to an existing system, product, or process. It details the nature of the proposed change, the reasons for the change, the potential impacts, and any associated risks. The document is typically reviewed and approved by relevant stakeholders before any changes are made. |
| Email Thread | An Email Thread is a chronologically ordered series of messages exchanged over the corporate mail system. Each message carries full header metadata (From, To, Cc, Date, Subject, Message-ID) and the complete body—often including quoted context from earlier replies. Because every revision and clarification is preserved verbatim, an email thread serves as a faithful, time-stamped record of decisions, technical justifications, and action items. |
| IM Thread | An Instant Messaging (IM) Thread captures a real-time chat conversation—typically from Slack, Microsoft Teams, or a similar platform—including timestamps, participants, and threaded replies. It preserves quick clarifications, decisions, and action items exchanged during day-to-day work. |
| FAQ / Q&A Sheet | An FAQ (Frequently Asked Questions) or Q&A Sheet provides answers to common questions, issues, or scenarios that are regularly encountered by teams or users. It organizes important clarifications, solutions, and best practices in a question-and-answer format, making it easy to refer to and address repeated inquiries. |
| API Reference / Interface Spec | An API Reference or Interface Specification document provides detailed information about the APIs (Application Programming Interfaces) or service endpoints that enable different software systems to communicate. This document includes the structure of API requests and responses, available methods, parameters, error handling, and any authentication or authorization requirements. |
| Release Notes / Changelog | Release Notes or Changelogs document the changes made in each version of a software product, including new features, enhancements, bug fixes, and any breaking changes. They help stakeholders, developers, and end-users track the progress and updates made to a system over time. |
| Test Plan / QA Checklist | A Test Plan or QA Checklist is a document that outlines the testing strategy for a new feature, system, or release. It defines the scope of testing, test cases, expected results, and the overall process to ensure that the software meets quality standards and is free from defects before it is released to production. |

Table 8: Common Enterprise Document Genres and Their Descriptions

Errors can also occur in the knowledge part. In some cases, if you want the model to generate a correct SQL in the specific areas, you need to supply the model with some specific knowledge. In Figure 12, the knowledge is the information that the model needs to gain from the retrieved documents. However, the model selected the wrong document library. Therefore, it did not obtain the correct knowledge, leading to subsequent SQL execution errors. This observation indicates that, beyond table retrieval, knowledge retrieval also poses a considerable challenge for the model.

In Figure 13, although the predicted SQL aligns with the gold SQL in terms of overall structure, it overlooks the critical rules regarding group references. This indicates that the model either neglects the relevant knowledge or exhibits deficiencies in its understanding of such knowledge.

**Question:**
For the patient who has the highest Ig A within the normal range, what is his or her diagnosis?

| Correct SQL: | Retrieved tables |
|---|---|
| SELECT patientData.patient_diag_clin_cond

FROM xelora1_thromb_patient AS patientData

INNER JOIN xelora1_thromb_lab_monitoring AS labData
  ON patientData.patient_exam_id = labData.patient_exam_id

WHERE labData.lab_iga_lvl BETWEEN 80
  AND 500 ORDER BY labData. lab_iga_lvl DESC LIMIT 1 | CREATE TABLE `xelora1_thromb_clin_exam` (
`patient_exam_id` INTEGER,
`exam_ts` DATE,
...
CREATE TABLE `apghi56_rdr_dg_workflow` (
`diagnostic_id` INTEGER,
`patient_id` INTEGER,
...
CREATE TABLE `apghi60_nibr_clin_diag` (
`diagnosis_id` TEXT,
`subject_id` TEXT,
... |
| **Predicted SQL:** | **Missed tables**
CREATE TABLE `xelora1_thromb_patient` (
`patient_diag_clin_cond` TEXT,
`patient_exam_id` TEXT,
...
CREATE TABLE `xelora1_thromb_lab_monitoring` (
`patient_exam_id` TEXT,
`lab_iga_lvl` TEXT,
... |
| SELECT patient_diag_clin_cond

FROM xelora1_thromb_clin_exam

WHERE exam_anti_cardiolipin_iga_conc <= 10

ORDER BY exam_anti_cardiolipin_iga_conc DESC

LIMIT 1 | |
| | **Documents** |
| | Omitted… |

Figure 10: An example for schema-retrieval error

**Question:**
What is the percentage of cards whose language is French among the Story Spotlight cards?

| Correct SQL: | Retrieved tables |
|---|---|
| SELECT CAST(SUM(CASE WHEN T2.
card_trans_lang = 'French'
THEN 1 ELSE 0 END) AS REAL) * 100 /
COUNT(T1.card_id) FROM
zentra11_mcard_crd_catalog AS T1
INNER JOIN zentra11_mcard_foreign_data AS T2
  ON T1.card_id_uuid = T2. card_id_ uuid
WHERE T1. card_attr_story_spotlight_ind = 1 | CREATE TABLE `zentra11_mcard_foreign_data` (
`card_id` INTEGER /*Examples: 1*/,
`card_id_uuid` TEXT,
`card_trans_lang` TEXT ,
...
CREATE TABLE `zentra11_mcard_crd_catalog_snapshot` (
`card_id` INTEGER,
`card_id_uuid` TEXT,
`card_attr_story_spotlight_ind` INTEGER,
... |
| **Predict SQL:** | CREATE TABLE `zentra11_mcard_crd_catalog` (
`card_id` INTEGER,
`card_id_uuid` TEXT,
`card_attr_story_spotlight_ind` INTEGER,
... |
| SELECT ROUND(100.0 * SUM(CASE WHEN
card_trans_lang = 'French' THEN 1 ELSE 0 END) /
COUNT(*), 2)
AS french_percentage
FROM zentra11_mcard_crd_catalog_snapshot
WHERE card_attr_story_spotlight_ind = 1; | |
| | **Documents** |
| | Omitted... |

Figure 11: An example for schema-linking error

**Question:**
How many drivers managed to finish the race in the 2008 Australian Grand Prix?

| Correct SQL: | Retrieved tables |
|---|---|
| SELECT COUNT(T2. driver_id)
FROM zentra13_f1mgr_races AS T1
INNER JOIN zentra13_f1mgr_drvr_race_results AS T2
ON T2. const_standings_race_id = T1. const_standings_race_id
WHERE T1. circuit_full_name = 'Australian Grand Prix'
AND T1. race_season_yr = 2008
AND T2. race_finish_tm IS NOT NULL | Omitted... |

| Predicted SQL: | **Documents** |
|---|---|
| FROM zentra13_f1mgr_drvr_race_results rr
JOIN zentra13_f1mgr_races r
ON rr.constructor_race_id = r.const_standings_race_id
WHERE r.race_season_yr = 2008
AND r.circuit_full_name = 'Australian Grand Prix'
AND rr.race_result_fin_pos IS NOT NULL | Document1: Formula 1 Driver Information Service API v1 Reference
...
### Field Specifications
**`racing_number`**: Integer representing the number permanently assigned to the driver during races. Maps to database field driver_racing_num.
...
Document2: F1 Database Schema Clarifications: Driver Laptimes & Standings Tables
From: Rajiv Desai
To: Data Engineering Team
...
Document3: Formula 1 Race Results Data: Frequently Asked Questions
...
Document4: Formula 1 Racing Data Management: Pit Stops & Results FAQ
...
Document5: Urgent: Clarifications Needed for F1 Reporting Schema
- - - - - - - - - - - - - - - - - - - - - - - - - - - - - - - - - - - - - - -
**Missed**
Target Document: RFC-2024-001:
The race_finish_tm field (finish time) uses different formats based on driver position: commonsense evidence: ;
1. if the value exists, it means the driver finished the race. |

Figure 12: An example for knowledge-retrieval error

**Question:**
Which group does superhero A-Bomb belong to?

| Correct SQL: | Retrieved tables |
|---|---|
| SELECT T2. superhero_race_classif
FROM zentos1_hrodom_char_metadata AS T1
INNER JOIN zentos1_hrodom_race AS T2
ON T1. sup_race_id = T2. align_id
WHERE T1.superhero_name = 'A-Bomb' | Omitted... |

| | **Documents** |
|---|---|
| **Predict SQL:**

SELECT ca.superhero_align_ethic_stance
FROM zentos1_hrodom_char_metadata AS cm
JOIN zentos1_hrodom_char_alignment AS ca
ON cm.superhero_align_id = ca.align_id
WHERE cm.superhero_id_name = 'A-Bomb' | Document1: Superhero Management System v2.0.0 Release Notes
...
## Overview
This major release introduces foundational changes to our superhero attribute
...
Document2: Superhero Registry API v1
Requests require HMAC-SHA256 signatures using client credentials issued by the Comic Industry Federation (CIF). Include headers:
...
Document3: Design Rationale
...
Document4: RFC-001: Design for Superhero Attribute Management System
...
Document5: RFC-001: Proposed Data Model Standardization for Superhero Attribute Management System
...
"commonsense evidence: In the context of superheroes, superhero_race_classif would refer to the particular group of people that the superhero belongs to base on these physical characteristics"
... |

Figure 13: An example for knowledge-grounding error

```
Prompt for Domain Expansion

# Task Description
Create an enterprise-grade **SQLite** database schema related to the topic **{TOPIC_SLOT}**.

# Requirements
1. The schema must contain **at least {MIN_TABLES_SLOT} tables** and at **least {MIN_COLUMNS_SLOT} columns** in total.
2. The schema must be substantially different from existing databases in both intended functionality and naming, while
still falling within the scope of the given topic. Do not duplicate or closely imitate any existing database. **Table
names must be unique: generating any table with the table name that duplicates one from the existing databases is
strictly prohibited.**
3. Provide only:
    - the database name;
    - for each table: its name, column names with data types, primary-key definition(s) and foreign-key constraint(s).
    - Do not include sample data.
4. Design the schema to reflect a realistic enterprise use case.
5. The schema you generate needs to be relevant to the topic; But it (including the names of database, tables and
columns) does not necessarily have to include keywords from the topic itself. For example, databases under the
"sports_and_athletes" topic can be related to any sport, such as tennis, baseball, chess, and so on. We encourage
generating **a diverse range of** databases that align with the topic.
6. The schema you generate must comply with the **SQLite database specifications**.
7. Your output should be entirely in English.

# Existing Databases (for exclusion check)
{EXSISTING_DATABASES_SLOT}

# Enterprise Schema Example
Here are some examples of enterprise schemas to help you understand the characteristics of enterprise database schemas,
Do not reuse its content or imitate its format:
```plain
'anefi_ods_yeb_asset_increase_order_delta_hh'(asset_order_id ''|gmt_create ''|gmt_modified ''|original_order_id ''|biz_request_id ''|user_id
''|biz_type ''|quotient ''|real_amount ''|status ''|biz_no ''|out_biz_no ''|bill_detail_id ''|biz_dt ''|trans_dt ''|pmt_dt ''|biz_context ''|memo
''|ext_info ''|contract_id ''|asset_account_type ''|asset_account_no ''|inst_id ''|sub_biz_type ''|cnl_pd_code ''|cnl_ev_code ''|cnl_no
''|biz_pd_code ''|biz_ev_code ''|pd_code ''|ev_code ''|payment_id ''|gmt_commit ''|business_type ''|fund_code 'Fund Code'|fund_inst 'Fund
Institution'|clear_dt 'Clearing Date'|dt ''|hour ")

'anods.ods_cfm_fund_order_delta_hh'(prefix 'Rowkey Prefix'|id 'Fund Document Number'|parent_id 'Parent Fund Document Number'|biz_trans_id
'Business Transaction ID'|biz_trans_code 'Business Transaction Code'|request_user_id 'Merchant Account ID'|original_from 'Business
Source'|access_channel 'Access Channel'|out_biz_no 'External Business Number'|order_no ' Order Number'|payment_no 'Payment Serial
Number'|sub_biz_type 'Sub-business Type'|payer_party_id 'Payer Participant ID'|payer_card_id 'Payer Card Info ID'|payer_card_no 'Payer Card
Number'|payee_party_id 'Payee Participant ID'|payee_card_id 'Payee Card Info ID'|payee_card_no 'Payee Card Number'|currency 'Currency
Code'|amount 'Transaction Amount'|real_amount 'Actual Amount'|remark 'Remark'|status 'Status'|sub_status 'Sub-status'|error_code 'Error
Code'|fail_reason 'Failure Reason'|memo 'Memo'|gmt_execute 'Execution Time'|gmt_create 'Creation Time'|gmt_modified 'Modification
Time'|gmt_expired 'Expiration Time'|gmt_confirm 'Confirmation Time'|gmt_pay 'Payment Time'|order_fee 'Order Pre-charge Amount (frozen
charge amount)'|charge_consult_no 'Pre-charge Document Number (charge currently 28 digits)'|cnl_pd_code 'Channel Product Code'|cnl_ev_code
'Channel Event Code'|cnl_no 'Channel Serial Number'|biz_pd_code 'Upstream Business Product Code'|biz_ev_code 'Upstream Business Event
Code'|pd_code 'Business Product Code'|ev_code 'Business Event Code'|ext_info 'Extended Field'|parent_biz_trans_id 'Parent Business Transaction
ID'|dt ''|hour ")
```

# Output Format
Return a single JSON object in your response and enclose it within <answer> and </answer>:

<answer>
{
    "db_name": "The name of the database you generated",
    "schema": {
        "the name of table 1": {
            "schema":{
                "the name of column 1": "data type of column 1",
                "the name of column 2": "data type of column 2",
                "the name of column 3": "data type of column 3",
                // other columns in table 1
            },
            "constraints":{
                "primary key": "PRIMARY KEY (`column x`)",
                "foreign keys": [
                    "FOREIGN KEY (`column x`) REFERENCES `table i` (`column y`)",
                    // other foreign key constraints
                ]
            }
        },
        "the name of table 2": {
            "schema":{
                // columns in table 2
            },
            "constraints":{
                "primary key": "primary key",
                "foreign keys" [
                    // foreign keys if exisit
                ]
            }
        },
        // other tables in this database
    }
}
</answer>

# Output
```

Figure 14: The prompt for domain expansion.

```
Prompt for Project and Area Name Generation

You are a senior enterprise database architect. You are helping a company restructure its legacy database into an
enterprise-grade system.

Database Name: {database_name}

You are given:
- Database name
- Database schema (DDL)
- database_description

Please:
1. Generate a project name (4-6 characters) for this database.The project name is randomly generated, has nothing to
do with the database, and does not repeat with the domain.
2. Generate a high degree domain abbreviation (3-8 characters) that captures the business domain

Output format:
Project Name: [your_project_name]
Domain Abbreviation: [your_domain_abbr]

[Database Schema]
{schema}
{desc_section}
```

Figure 15: The prompt for table project and area name generation.

```
Prompt for Enterprise Table Name Generation

You are an enterprise database architect helping rename a table to follow enterprise naming conventions.

Please rename the table {table_name} using the following pattern:

Naming Pattern:
<project>_<domain>_<content>

Where:
- <project>: {project} (already determined)
- <domain>: {domain} (already determined)
- <content>: Semantic summary of the table's core entity or process

Example:
Original table: user_behavior
Expected: {project}_{domain}_usr_behavior

Now rename:

Table name: {table_name}

Guidelines:
1. Use precise, business-aware terms for <content>
2. Be concise but descriptive
3. Use common abbreviations (usr for user, txn for transaction, etc.)
4. Select the most appropriate partition suffix from the provided options

Only output the new table name as a single line. Do not include any explanations or formatting.
```

Figure 16: The prompt for table name generation.

```
┌─────────────────────────────────────────────────────────────────────────────┐
│ Prompt for Area Explanation                                                   │
├─────────────────────────────────────────────────────────────────────────────┤
│                                                                               │
│ You are a senior enterprise database architect. You are helping a company restructure its legacy database into an │
│ enterprise-grade system.                                                      │
│                                                                               │
│ You are provided with the following:                                          │
│ - A relational database schema (DDL format);                                  │
│ - A column-level description file (if available), which includes: original column names, semantic names, textual │
│ descriptions, and possible value explanations.                                │
│                                                                               │
│ Your task is to perform a domain-level, enterprise-grade analysis of the database and return a structured summary for │
│ downstream usage.                                                             │
│                                                                               │
│ ## Your output should include the following sections:                         │
│                                                                               │
│ ### 1. Business Domain Classification                                         │
│ - Clearly identify the high-level business domain the database belongs to.    │
│ - Provide one or more enterprise-level subdomains involved                    │
│ - Use concise and enterprise-recognizable terminology.                        │
│                                                                               │
│ ### 2. Table Themes and Functional Definitions                                │
│ - For each table in the schema, explain its business purpose.                 │
│ - Describe the type of entity or process the table models                     │
│                                                                               │
│ ### 3. Field Classifications and Semantic Grouping                            │
│ - Organize fields into functional groups, such as:                            │
│ - Identification fields - Address/location fields                             │
│ - Operational attributes                                                      │
│ - Classification codes                                                        │
│ - For each group, summarize its business purpose and usage context.           │
│                                                                               │
│ ### 4. Semantic Repair and Enrichment                                         │
│ - Identify any columns whose descriptions are missing, vague, or purely abbreviations. │
│ - For such fields, infer and supplement accurate definitions based on domain knowledge. │
│ - Emphasize clarity and completeness over brevity. Avoid unexplained abbreviations unless they are domain standards │
│ and include their expansion.                                                  │
│                                                                               │
│ ### Formatting Constraints:                                                   │
│ - Be concise but informative.                                                 │
│ - Avoid repeating the schema line-by-line; instead, provide a business-level summary. │
│ - Use bullet points or short paragraphs for clarity.                          │
│ - Do not assume a single fixed domain.                                        │
│                                                                               │
│ Expected Output Format:                                                       │
│                                                                               │
│ Business Domain Classification:                                               │
│ This database primarily serves the following domain(s):                       │
│ - [High-level domain, e.g., Public Education, Financial Services, Healthcare...] │
│                                                                               │
│ Enterprise Subdomains:                                                        │
│ - [Subdomain A]: [Short description, e.g., "Manages the lifecycle and attributes of registered educational │
│ institutions."]                                                               │
│ - [Subdomain B]: [Short description, e.g., "Tracks charter classification and operational status of schools."] │
│ ...                                                                           │
│                                                                               │
│ Table Classification and Definitions:                                         │
│ - table_name_1: [Concise description of what entity or process the table represents and its business purpose.] │
│ - table_name_2: [Concise description...]                                       │
│ ...                                                                           │
│                                                                               │
│ Field Classification and Groupings:                                           │
│ - Identification Fields:                                                       │
│   - field_a: [Business meaning]                                               │
│   - field_b: ...                                                              │
│ - Geolocation / Address Fields:                                               │
│   - ...                                                                        │
│ - Status / Operational Attributes:                                            │
│   - ...                                                                        │
│ - Classification / Typing Codes:                                              │
│   - ...                                                                        │
│ ...                                                                           │
│                                                                               │
│ Semantic Repair and Enrichment:                                               │
│ - [field_name_1]: [Inferred or repaired explanation. Expand abbreviation if unclear, and describe usage.] │
│ - [field_name_2]: ...                                                         │
│ ...                                                                           │
│                                                                               │
│ **Database Schema:**                                                          │
│ {schema_ddl}                                                                  │
│ {desc_section}                                                               │
│                                                                               │
└─────────────────────────────────────────────────────────────────────────────┘
```

Figure 17: The prompt for column name area explanation.

---

### Prompt for Column Names Expansion

You are an enterprise database architect specializing in field expansion and semantic enrichment. Your task is to transform academic or legacy column names into enterprise-grade naming patterns.

Enterprise Naming Recommendations:
1. Avoid using casual prefixes like is, has, or was. Use neutral, abstract, and semantically-rich terms instead.
2. Ensure column names reflect business meaning, embedding semantics and context without redundancy.
3. Supplement the column name with contextual information about the table and related fields. The contextual information needs to conform to the column name logic and structured naming.

Domain Adaptation Rule:
- The database you are working with may belong to a specific business domain.
- If a certain column concept or terminology is typically expressed using domain-specific naming conventions, please adapt to that convention naturally.
### Domain Context:
{domain_analysis}
Instructions:
1: Suggest a full, enterprise-style column name that clearly expresses its business meaning. You may rewrite or reorder components, but do not omit or change key context words already in the original name.
2: If a more commonly used business domain synonym exists for a component, perform the replacement. Only replace terms that do not affect the core semantic scope.
3: Ensure that the column name covers both the column description and value meanings in a meaningful but non-redundant way.

### Column Information:
- Table: {column_info.table_name}
- Original Name: {column_info.original_name}
- Description: {description_display}
- Data Type: {column_info.data_format}
- Value Semantics: {value_desc_display}
{sample_text}

Final Output Format:
### Final Column Name: ...

Your response must include the exact header '### Final Column Name:'

Figure 18: The prompt for column name expansion.

---

### Prompt for Column Name Abbreviation

You are an enterprise database engineer. Your task is to abbreviate and normalized the following enterprise-style column name.
Unify similar rules:
for any synonymous or overlapping terms in the input,
choose one canonical wording and apply it consistently across all outputs.
Prefer clearly abbreviated words over ad-hoc name.
For each category of indicators, choose one consistent naming convention within the same table. For example, for level-type indicators, always use either level or lvl consistently. For countable indicators, unify choices among cnt, count, or total and apply the same form to all related columns. Apply this principle to other types of indicators as well.

Abbreviate Rules:
1. Use snake_case format only.
The abbreviation rule for snake case is to determine whether each component between underscores can be abbreviated.
Example:
description_product_number: desc_prod_num
2. Retain the integrity of the meaning of the core terms in the Original Name and ensure that experts in the relevant field can understand its meaning after abbreviation.
Do NOT use single-letter initials.
3.If a token appears multiple times within the same output batch, always abbreviate it consistently.
4. Abbreviations should be as close to the original meaning as possible—easy to understand and semantically clear. If you think a word is already concise, you don't need to abbreviate it.

### Column Information:
- Full Enterprise Name in the table: {full_column_names}

### Output Format:

Final Column Names in the table: [Final names]

Your response must include the exact headers shown above.

Figure 19: The prompt for column name abbreviation.

1404
1405
1406
1407
1408
1409
1410
1411
1412
1413
1414
1415
1416
1417
1418
1419
1420
1421
1422
1423
1424
1425
1426
1427
1428
1429
1430
1431
1432
1433
1434
1435
1436
1437
1438
1439
1440
1441
1442
1443
1444
1445
1446
1447
1448
1449
1450
1451
1452
1453
1454
1455
1456
1457

## Prompt for Document Generation

# Background
To build a benchmark that truly reflects **enterprise-grade Text-to-SQL** scenarios, we must replicate how knowledge is distributed inside real companies:
- Business / domain knowledge is scattered across meeting minutes, change requests, ops runbooks, emails, and other heterogeneous documents.
- A querying system must **retrieve** information from a vast collection of external documents **before** generating SQL.

Your task is to **sparsify** the external knowledge you receive—turning it into a single Markdown document that resembles a real-world knowledge artifact.

# Task Description
You are given two sources of external knowledge:
1. **Database description** – full table/column names, value explanations, and embedded business / domain knowledge.
2. **Extra knowledge** – additional domain knowledge not present in the description file.

**Your goal is to transform this input into one Markdown document of at least {MIN_WORDS_SLOT} words, written in the specified genre as detailed below**:
{DOC_GENRE_SLOT}

# Requirements
1. **No knowledge may be omitted.**
   - Every piece of original external knowledge **must appear somewhere in the document, except for `sample_value` and `data_format`**.
     *This explicitly includes:*
     * All rows from the database description table (Including 3 columns: `original_column_name`, `column_description`, `value_description`),
     * every item contained in the extra-knowledge input.
   - If any information cannot be woven naturally into the narrative, list it verbatim in a "① Loose Notes" block at the end of the document—*otherwise do not include that block*.

2. **Tag original external knowledge for manual completeness checks. Use the tagging scheme below:**
   - For each row *i* (starting at 1) in the database-description table, tag each non-empty field as:
     - `original_column_name` → <ocn{i}> … </ocn{i}>
     - `column_description`    → <cd{i}> … </cd{i}>
     - `value_description`     → <vd{i}> … </vd{i}>
   - The `sample_value` and `data_format` columns are provided for your reference to help you understand the external knowledge. You **may include relevant content** in the final document if appropriate, but **must not apply any tags** to it.
   - If a field in that row is blank, **omit the corresponding tag; never invent content**.
   - For each extra-knowledge item *k* (in provided order), tag it as <ek{k}> … </ek{k}>.
   - The text inside every tag must come directly from the original external knowledge (verbatim or with only trivial rephrasing that leaves the meaning unchanged).
   - Distribute tagged elements throughout the document and do not cluster them in a single section if possible. Avoid mechanically listing tags and the knowledge within tags (e.g., directly providing a list).

3. **Sparsification & contextualization**
   - **Do not** present a tidy data-dictionary list such as "table-field-meaning".
   - Knowledge should be woven naturally into the chosen genre and dispersed across the document.
   - You may appropriately expand upon the original external knowledge to make the document more natural and realistic. However, the final document **must not** contain any content that contradicts the original external knowledge.
   - If new columns that are not present in the description file are mentioned (e.g., due to simulated discussions or change proposals), they must be framed negatively—for example, discussing their removal or disagreement over adding them.

4. **Directory path**
   - Provide a relative path for the document (e.g., `design/RFC-0421.md` or `ops/playbook/alert_CPU.md`) to show hierarchy.

5. **Independent theme**
   - The document's subject matter **must not** directly reference the task itself (e.g., Text-to-SQL benchmark, enterprise Text-to-SQL, External Knowledge). Choose varied, realistic business or technical topics instead.

# Output Format
Return a single JSON object in your response and enclose it within <answer> and </answer>:

<answer>
{
  "path": "your/path/here",
  "genre": "{DOC_GENRE_NAME_SLOT}", // Only the genre name needs to be given
  "title": "the title of the document",
  "content_md": "the content of the document"
}
</answer>

# Original External Knowledge
## Database description
{DB_DESCRIPTION_SLOT}

## Extra knowledge
{EXTRA_KNOWLEDGE_SLOT}

# Output

Figure 20: The prompt for enterprise knowledge document generation.

```
Prompt for Doucument Check & Correct

# Background
In the prior task, your colleague has taken concise, densely packed external knowledge and—following the
sparsification & tagging rules—expanded it into **one Markdown document** written in the specified genre. This
artifact will later feed a retrieval-based Text-to-SQL benchmark, so its accuracy is critical.

## Prior task description and materials
### Prior task description
You are given two sources of external knowledge:
1. **Database description** – full table/column names, value explanations, and embedded business / domain
knowledge.
2. **Extra knowledge** – additional domain knowledge not present in the description file.

Your task is to **sparsify** the external knowledge you receive—turning it into a single Markdown document
that resembles a real-world knowledge artifact.

### Database description
{DB_DESCRIPTION_SLOT}

### Extra knowledge
{EXTRA_KNOWLEDGE_SLOT}

### Target genre
{DOC_GENRE_SLOT}

### **Sparsification & tagging rules**
   - **No knowledge may be omitted.**
      - Every piece of original external knowledge **must appear somewhere in the document, except for
`sample_value` and `data_format`**.
……

(Omitted here, you can refer to the same part of the pormpt for document generation)

……
## Candidate document
To assist you in identifying errors, the candidate document has already been pre-checked according to specific
rules, and any detected issues have been marked accordingly. (Note: The rule-based check is limited in
capability and may miss some errors — you must carefully identify any remaining issues during your review.)
```md
{CANDIDATE_DOC_SLOT}
```

# Your Task
Audit the candidate document against the source materials and the rules, then correct it where necessary.

## Checklist
1. **Completeness**
   - Every element of the original external knowledge except for `sample_value` and `data_format` must appear
in the document—either in the main text or, if it cannot be woven in naturally, in a "① Loose Notes" block.

2. **Tag validation & correction**
   - **Structural correctness** Each tag must have a valid index, a recognized tag type (ocn, cd, df, vd, ek),
and must appear in proper pairs (<tag> and </tag>). Fix any malformed or unmatched tags.
   - **Existence** If the corresponding field in the database description is blank, delete the tag **and its
enclosed content**.
   - **Accuracy** The text inside every tag must match the source verbatim, or with only trivial re-phrasing
that preserves meaning. Replace mismatches with the correct source text.

3. Check whether any other part of the document violates the sparsification and tagging rules. If such
violations are found, revise the document accordingly to ensure full compliance.

## Requirements
1. Modify **only** the content that violates the checklist or rules; everything else must remain unchanged.
2. You only need to revise the document content – no need to provide the title, genre, or file path.
3. When checking for completeness, go through the database description row by row, verifying each column field
individually. Then do the same for each item in the extra knowledge section. Present the results as a
structured checklist.
4. When validating tags, iterate over each tag that appears in the document, and check the following
dimensions: Structural correctness, Existence, Accuracy. Present the tag validation as a checklist as well.

# Output format
Provide **one final, corrected Markdown document** that meets all requirements, and wrap it exactly as shown:
<answer> ...your corrected Markdown... </answer>

# Output
```

Figure 21: The prompt for checking and correcting enterprise knowledge document.

## Prompt for Evaluation on BIRD-Ent

```
# Task Description
You own a data asset (similar to a data warehouse) that has collected a series of tables from different sources around
a specific topic. You are now given the schemas and data sources for all the tables within this data asset. To help
you better answer the question, your colleague has pre-selected several tables that may be relevant to the question.
Based on this information, please write an SQLite query to answer the following question.

# Requirements:
1. You only need to provide one complete, correct, and executable SQL statement.
2. Do not include any additional information outside of the SQL query in your response.

# Output Format
Your SQL should be placed between <answer> and </answer>, as follows:
<answer>your SQL query</answer>

# Data Table Naming Standards
Below are the naming conventions for the data tables, which will help you understand the following Data Asset
Information:
Table naming conventions: <project>_<area>_<content>.
- <project>: The project/product or data domain identifier to distinguish teams, business lines, or application
contexts.
- <area>: Business domain and subject area identifier which the table belongs.
- <content>: The summary of the table's content.

# Data Asset Information
{SCHEMA_SLOT}

# External Knowledge
The following document may contain useful information needed to answer the question and is provided for your reference:
{KNOWLEDGE_SLOT}

# Output
Question: {QUESTION_SLOT}
Your Answer:
```

Figure 22: The prompt for evaluation on the BIRD-Ent Benchmark.

## Prompt for Evaluation on Spider-Ent

```
# Task Description
You own a data asset (similar to a data warehouse) that has collected a series of
tables from different sources around a specific topic. You are now given the schemas
and data sources for all the tables within this data asset. To help you better answer
the question, your colleague has pre-selected several tables that may be relevant to
the question. Based on this information, please write an SQLite query to answer the
following question.

# Requirements:
1. You only need to provide one complete, correct, and executable SQL statement.
2. Do not include any additional information outside of the SQL query in your
response.

# Output Format
Your SQL should be placed between <answer> and </answer>, as follows:
<answer>your SQL query</answer>

# Data Asset Information
{SCHEMA_SLOT}

# Output
Question: {QUESTION_SLOT}
Your Answer:
```

Figure 23: The prompt for evaluation on the Spider-Ent Benchmark.

