# OpenReview forum: "Text-to-SQL Benchmarks for Enterprise Realities: Under Massive Scopes, Complex Schemas and Scattered Knowledge"
_ICLR.cc/2026/Conference — ICLR 2026 Conference Withdrawn Submission_

### Official Review · Reviewer_mCRi · 2025-10-26

**Soundness:** 2
**Presentation:** 3
**Contribution:** 1
**Rating:** 2
**Confidence:** 4

**Summary:**

This paper targets the gap between academic Text-to-SQL benchmarks and real-world enterprise scenarios, which require handling massive query scopes, complex schemas, and scattered knowledge across large document collections.
The authors propose a cost-effective refinement framework that transforms existing benchmarks (BIRD and Spider) into enterprise-grade benchmarks (BIRD-Ent and Spider-Ent) through domain, schema and knowledge refinement, alongside introducing the DRAG Text-to-SQL paradigm that explicitly models retrieval before generation. Evaluation on state-of-the-art LLMs reveals substantial performance drops to 39.1% on BIRD-Ent and 60.5% on Spider-Ent, demonstrating that current systems struggle with enterprise-level complexity.

**Strengths:**

1. The paper targets an important and well-motivated problem that there is still a critical gap between academic Text-to-SQL benchmarks and enterprise requirements across dimensions such as scope, schema and required knowledge.

2. The benchmark refinement idea provides a scalable approach to creating realistic benchmarks without expensive enterprise data collection, addressing privacy and annotation cost barriers.

**Weaknesses:**

1. Vague enterprise challenge description without real-world validation. The paper claims to capture "enterprise realities" but provides no analysis or validation of what real enterprise data actually looks like. While Figure 4 shows one anonymized example, there is no quantitative comparison of characteristics (e.g., schema complexity metrics, knowledge distribution patterns, naming convention statistics) between real enterprise databases and the synthetic benchmarks. Without such validation, it remains unclear whether the identified challenges (massive scope, complex schemas, scattered knowledge) accurately reflect the most critical pain points in real deployments versus BEAVER, which use actual enterprise data.


2. The refinement framework makes several assumptions that also lack empirical grounding: (1) Domain expansion uses LLMs to generate synthetic tables at scale, but there is no evidence these tables exhibit realistic statistical properties or semantic relationships found in real enterprise databases; (2) Schema-level refinement assumes enterprises follow clean hierarchical naming conventions, but real enterprise schemas are often messier with inconsistent conventions, legacy naming, and organic evolution; (3) Knowledge-level refinement assumes well-documented, enterprise-style documents, but real enterprises often have incomplete, outdated, or poorly maintained documentation. These design choices may create artificial rather than authentic enterprise challenges.


3 Insufficient comparison with existing enterprise benchmarks. Table 1 claims the benchmarks achieve higher "Enterprise Realism" than BEAVER and Spider 2.0, but this rating is subjective without supporting analysis. The paper does not provide: (1) direct performance comparison on the same models between Ent-series and BEAVER/Spider 2.0 to assess relative difficulty; (2) quantitative comparison of dataset characteristics (beyond simple statistics like table/column counts) to justify why synthetic data better captures enterprise complexity than real data; (3) analysis of whether the challenges introduced (retrieval difficulty, schema noise, knowledge scattering) align with where real systems fail in practice. BEAVER explicitly uses real enterprise data and highlights similar challenges, the paper needs stronger justification for why synthetic refinement is superior or complementary.

**Questions:**

see weakness

---

> ### Author Response · Authors · 2025-12-03
>
> We sincerely appreciate your recognition of our work and your valuable feedback. Below, we provide our responses and clarifications to the weaknesses and questions you have raised
> ## W1) Systematic Validation of Enterprise Realism
>
> We ensure the *enterprise realism* of our benchmark from multiple perspectives, including analyses and validations grounded in real-world evidence.
>
> 1. **Statistical analysis over 324 enterprise data asset domains and multiple benchmarks.**
>
>    - We sampled 2,000 tables from real enterprise data assets to analyze enterprise-specific naming characteristics, including long names, hierarchical naming, extensive abbreviations, and domain-specific terminology:
>
>      |                           | Real Enterprise | Spider-Ent | BIRD-Ent | BEAVER | Spider 2.0-snow/lite | Spider 2.0-DBT | Spider | BIRD |
>      |---------------------------|------------------|------------|----------|--------|-----------------------|----------------|--------|------|
>      | Avg. Table name length (tokens) | **10.2** | 5.2 | 5.2 | 2.6 | 8.3 | 3.3 | 1.4 | 1.2 |
>      | Hierarchical naming       | **Y**            | Y | Y | N | Y | N | N | N |
>      | Avg. Column name length (tokens) | **3.5** | 2.1 | 2.1 | 2.1 | 3.75 | 2.3 | 1.8 | 1.9 |
>      | Avg. token Zipf freq.     | **3.92** | 4.25 | 4.25 | 4.55 | 4.71 | 4.55 | 4.91 | 4.66 |
>
>      `Avg. token Zipf freq.` is computed by splitting all table and column names into tokens, obtaining the Zipf frequency of each token using the *wordfreq* library (higher Zipf values indicate more common English words), and then averaging these values across the dataset. Both real enterprise data and our Ent-series benchmarks show relatively low values on this metric, because real enterprise schemas contain a large number of abbreviations and domain-specific terms.
>
>    - We also analyzed the query scope of real enterprises:
>
>      |                     | Real Enterprise | BIRD-Ent | Spider 2.0-snow/lite | Others |
>      |---------------------|------------------|----------|----------------------|--------|
>      | # Table / QS        | 625.8            | 392.1    | 50.4                 | See Table 1 in the paper for more details. |
>
> 2. **Expert consultation and blind evaluation.**
>
>    During the preliminary investigation phase before formally constructing the benchmark, we conducted in-depth discussions with multiple enterprise experts and collected a large number of real-world use cases. Our characterization of enterprise scenarios was recognized by these experts and validated against concrete examples.
>
>    After the benchmarks were constructed, we further invited 4 enterprise experts to conduct blind evaluations on multiple benchmarks. The evaluation considers seven dimensions: task setting, schema scale, schema style, knowledge scale, knowledge content, natural language questions, and SQL queries. The benchmarks were ranked according to their closeness to real enterprise scenarios. The final results are shown below:
>
>    | Metric | Spider-Ent | BIRD-Ent | BEAVER | Spider 2.0-lite/snow | Spider 2.0-DBT | Spider | BIRD |
>    |--------|------------|----------|--------|-----------------------|----------------|--------|------|
>    | Average Rank Score | 3.1 | 1.6 | 3.4 | 2.0 | 2.6 | 4.3 | 3.6 |
>    | Final Rank | 4 | 1 | 5 | 2 | 3 | 7 | 6 |
>
> 3. **Support from authoritative references.**
>
>    - Recent work (DOI: 10.18653/v1/2025.emnlp-main.51, arXiv: 2507.23104) discusses the large-scale schema issue commonly observed in real-world and enterprise databases.
>    - Official documents from Oracle and Alibaba Cloud
>      (https://docs.oracle.com/cd/E38317_01/doc.11117/e37988/namingconv.htm,
>      https://www.alibabacloud.com/help/en/dataworks/user-guide/data-warehouse-layering)
>      recommend or directly adopt hierarchical table naming conventions similar to those used in our benchmark.
>    - Documentation from SAP BW Consulting
>      (https://www.sapbwconsulting.com/blog/data-warehouse-naming-conventions-best-practices-for-sap)
>      indicates that enterprise data warehouses follow column and table naming principles similar to ours, including richer business semantics and finer granularity.
>    - A recent study (DOI: 10.18653/v1/2025.findings-emnlp.1348) reports extensive use of abbreviations in real enterprise table and column names, consistent with our benchmark design.
>    - Another work (DOI: 10.18653/v1/2025.acl-demo.27) highlights the existence of legacy tables caused by version evolution in real-world databases, corresponding to the redundant tables in our benchmark.
>
> 4. **Industry consensus.**
>
>    Massive query scope, complex database schemas, and knowledge scattered across documents are natural data characteristics of large-scale enterprise data systems, arising from cross-system integration, data life cycles, and non-standardized processes.

---

> ### Author Response · Authors · 2025-12-03
>
> ## W2) Assumptions Behind the Refinement Framework
>
> 1. As shown in W1, we provide a systematic analysis of enterprise characteristics. From a statistical perspective, our benchmark is closer to real enterprise databases. For the focus of our benchmark, **semantic relations are not the primary concern**, because we target **structural enterprise challenges** (i.e., how to handle large-scale and complex schemas), rather than the understanding of specific business content (we do not evaluate whether models understand enterprise business logic).
>
> 2. The assumption that *“enterprises follow clean hierarchical naming conventions”* is **not unfounded**:
>    - The notion of “clean naming” is relative to *non-enterprise data*. Due to the massive scale of data assets, enterprises inevitably adopt standardized naming conventions to facilitate data integration. Official documentation from Oracle and Alibaba Cloud
>      (https://docs.oracle.com/cd/E38317_01/doc.11117/e37988/namingconv.htm,
>      https://www.alibabacloud.com/help/en/dataworks/user-guide/data-warehouse-layering)
>      explicitly recommends or directly adopts hierarchical table naming conventions similar to those used in our work.
>    - When designing schema-level refinement, we also take **“messy” factors** into account, such as introducing redundant tables and columns caused by historical legacy and business evolution, as well as increasing the abbreviation level of column names.
>    - Even though it is impossible to cover all possible sources of “messiness,” our benchmark already moves **one step closer to enterprise reality** compared to existing benchmarks. Even BEAVER and Spider 2.0 only use partial data from a few companies and cannot cover the full spectrum of enterprise scenarios.
>
> 3. Compared with existing benchmarks—which **do not consider knowledge scattered across documents at all**—our benchmark has already made a **substantial step toward enterprise-level realism**.
>
> ---
>
> ## W3) Quantitative Comparison with Existing Enterprise Benchmarks
>
> First, as discussed in W1, we validate the *enterprise realism* of our benchmark from multiple perspectives.
>
> Second, we emphasize that our contribution lies in **identifying and amplifying the three enterprise challenges discussed in the paper**. These challenges are well documented in industry practice (see W1.1), yet have not been sufficiently reflected in existing benchmarks. Enterprise Text-to-SQL ultimately requires a **diversified benchmark ecosystem**, and our goal is to **complement this ecosystem**, rather than to construct a benchmark that is simply more difficult than existing ones (indeed, harder does not necessarily mean better).
>
> We now respond point by point to the aspects you believe are missing:
>
> 1. **On relative difficulty evaluation**:
>    The purpose of our benchmark is **not to deliberately increase difficulty**, but to realistically reflect the key challenges faced by Text-to-SQL in enterprise settings. According to our experiments in the paper, our benchmark indeed poses substantial challenges to existing retrieval methods and LLMs. Moreover, since our task setup differs fundamentally from that of BEAVER and Spider 2.0, **direct score-based difficulty comparisons are not meaningful**.
>
> 2. **On quantitative dataset comparison and enterprise complexity validation**:
>    As discussed in W1, we support our claims about the three major enterprise challenges through **quantitative analysis, expert validation, and literature references**. Although statistics such as the number of tables and columns are simple, they are also the **most direct indicators** of the challenges we target. We have never claimed that our synthetic data are *more complex than real enterprise data*; rather, we aim to **expose and model specific challenges inherent in real enterprise scenarios**.
>
> 3. **On whether the identified challenges align with real-world failure modes**:
>    As discussed in W1, our challenge identification is derived directly from **enterprise experts, real enterprise use cases, publicly available enterprise documents, and industry consensus**.

---

### Official Review · Reviewer_i4ET · 2025-10-28

**Soundness:** 3
**Presentation:** 3
**Contribution:** 3
**Rating:** 6
**Confidence:** 4

**Summary:**

This paper introduces BIRD-Ent and Spider-Ent, two benchmarks simulating enterprise Text-to-SQL by refining existing benchmarks. They expand query scopes to 4,000+ columns (55-165× larger), add enterprise-style schemas, and scatter knowledge across 1.5M tokens of documents. Models must retrieve tables and knowledge before generating SQL. SOTA LLMs drop from 77.5%→39.1% on BIRD and 91.2%→60.5% on Spider, revealing that current models aren't enterprise-ready.

**Strengths:**

1. Important problem: Enterprise Text-to-SQL is different from academic settings, and we need benchmarks that reflect this. The paper makes this gap concrete with clear evidence.
2 .Interesting generation approach: Using LLMs to refine academic benchmarks is practical and clever. The document generation pipeline (segmenting knowledge, choosing enterprise genres like RFCs/FAQs, embedding with tags, then self-correction) is well-designed. Creating 1,412 realistic documents at 1.5M tokens would be prohibitive manually.
3. Clear empirical findings: The performance drops are large and consistent across models. The error analysis shows retrieval is the main bottleneck (knowledge retrieval only 14.9% perfect recall). Even with perfect retrieval, models only hit 51.8%, showing the problem isn't just retrieval.

**Weaknesses:**

1. Cost claims don't match reality: The paper claims "very low cost" and "minimal human intervention" but Spider-Ent needed 100% manual review with 40-60% rejection. BIRD-Ent needed 10% double annotation. This is substantial work.

2. DRAG isn't novel: BEAVER already does table retrieval. Adding knowledge retrieval is the obvious next step, not a paradigm shift. The real contribution is the benchmark scale and empirical findings, not inventing dual retrieval. The framing oversells this.

**Questions:**

1. What are the actual costs? Person-hours for annotation? LLM API costs? How does this compare to building BIRD/Spider from scratch? Numbers would help justify the cost-effectiveness claim.

2. Why did Spider-Ent have such high rejection? 40-60% failure rate suggests the refinement approach has limits. Is this a problem with Spider's underspecification or your method?

---

> ### Author Response · Authors · 2025-12-03
>
> We sincerely appreciate your recognition of our work and your valuable feedback. Below, we provide our responses and clarifications to the weaknesses and questions you have raised.
> ## W1) Annotation Cost
>
> Although we incurred a certain amount of cost for manual verification, the **human cost for dataset annotation itself** (including database collection, natural language question writing, and SQL annotation) is **almost zero**, which is typically the most labor-intensive and expensive stage of benchmark construction.
>
> To make our claim more convincing, we report the actual annotation cost and compare it with **Spider 1.0** (other benchmarks do not disclose their human annotation costs):
>
> | Benchmark     | Task                  | Questions | Cost/Item (person-hour) | Total Cost (person-hour) |
> |---------------|------------------------|-----------|--------------------------|---------------------------|
> | **BIRD-Ent**  | Answer uniqueness      | 153       | 0.17                     | 26.0                      |
> |               | Semantic alignment     | 153       | 0.07                     | 10.7                      |
> |               | Document correctness   | 141       | 0.10                     | 14.1                      |
> |               | **Total**              | -         | -                        | **50.8**                  |
> |               |                        |           |                          |                           |
> | **Spider-Ent**| Answer uniqueness      | 1034      | 0.10                     | 103.4                     |
> |               | Semantic alignment     | 103       | 0.05                     | 5.2                       |
> |               | Document correctness   | -         | -                        | -                         |
> |               | **Total**              | -         | -                        | **108.6**                 |
> |               |                        |           |                          |                           |
> | **Spider 1.0**| **Total**              | -         | -                        | **1100.0**                |
>
> These results show that the **human cost of our benchmark is significantly lower than that of Spider 1.0**. Other benchmarks, like Spider 1.0, also rely on fully manual annotation and verification. Therefore, the human cost of Spider 1.0 is representative to a certain extent.
>
> ---
>
> ## W2) Novelty of the Paradigm
>
> We agree that **knowledge retrieval can be viewed as a natural extension of table retrieval**, yet **existing Text-to-SQL research does not mention or study this step at all**. Therefore, our introduction of this component is indeed pioneering.
> Our results further show that in real enterprise settings, **knowledge retrieval is indispensable**, and DRAG provides the **first modular and measurable formulation of this previously overlooked requirement**.
>
> ---
>
> ## Q1) Actual Annotation Cost and Comparison
>
> See **W1**.
>
> ---
>
> ## Q2) Reason for the High Rejection Rate of Spider-Ent
>
> The high rejection rate of Spider-Ent is mainly due to the **vagueness of natural language questions in Spider** and the presence of many **semantically overlapping databases**.
>
> For example, for the question *“What is the total number of singers?”*, the Spider dataset contains multiple databases with very similar content, such as *concert_singer*, *singer*, and *music_1*, all of which include the *singer* entity. It is therefore impossible to determine which database the question refers to, resulting in **non-unique answers**.
>
> In contrast, such direct duplication of databases is rare in **BIRD**, and the question references are generally much clearer.

---

### Official Review · Reviewer_HwqY · 2025-10-29

**Soundness:** 2
**Presentation:** 2
**Contribution:** 2
**Rating:** 2
**Confidence:** 5

**Summary:**

This paper introduces new Text-to-SQL benchmarks, BIRD-Ent and Spider-Ent, designed to better reflect enterprise realities. These benchmarks expand the schema size to thousands of columns, inject naming noise, and simulate document-based external knowledge retrieval. They are built by refining existing benchmarks (BIRD and Spider) using LLMs to generate additional tables, schema variations, and embed relevant knowledge within lengthy document collections. The paper also defines the DRAG (Dual-Retrieval-Augmented-Generation) formulation, where both table schemas and knowledge documents must be retrieved before generating SQL. Evaluation shows that many LLMs perform worse on these new benchmarks.

**Strengths:**

1. The paper addresses an important problem: building Text-to-SQL benchmarks that more accurately reflect realistic, enterprise-style scenarios.

2. It provides a detailed error analysis, categorizing failure modes such as schema retrieval, knowledge grounding, and SQL construction, which helps identify where current models fall short.

3. The benchmark is evaluated across a diverse set of models, including both proprietary (e.g., GPT-4o) and open-source (e.g., Qwen3, DeepSeek) LLMs of varying sizes, offering a comprehensive view of model performance.

**Weaknesses:**

1. Concerns about quality control. The benchmark relies heavily on LLM-generated artifacts, including additional tables, document synthesis, and schema perturbations. In such a setting, quality control is crucial. However, the current procedure appears limited: only 10% of the data is manually inspected with double annotation, and details on the audit process are sparse. It remains unclear how key properties are verified. For example, how “answer uniqueness” is ensured when LLMs might generate semantically similar or redundant tables, or how the “enterprise-style” nature of generated documents and column names is assessed. Without more transparency and rigor, the reliability of the benchmark is difficult to evaluate.
2. Vague definitions of enterprise characteristics: Several key terms, such as “enterprise-style schemas” or “enterprise knowledge documents”, are described in vague and informal ways. The paper lacks a clear and systematic abstraction of what constitutes an enterprise setting. For instance, the authors mention converting original table names into hierarchical formats like "project area content", but this design choice appears ad hoc. Moreover, the paper narrowly focuses on retrieval challenges, whereas enterprise settings may also involve more complex user questions that require sophisticated reasoning and compositional SQL generation, an aspect that is underexplored here.
3. Limited evaluation scope: The experimental evaluation is limited to vanilla LLMs, without considering existing SOTA Text-to-SQL systems that already integrate retrieval capabilities. Many recent methods, such as CHASE-SQL and AskData (https://arxiv.org/abs/2505.19988), incorporate schema and knowledge retrieval as part of their pipelines. More recent RL-based methods (e.g., arXiv:2503.23157, arXiv:2503.00223) even bake retrieval and schema linking directly into the model. It remains unclear whether these more capable models also fail on this benchmark. For example, if an LLM is paired with a keyword search tool, would it still struggle under the proposed setting? Without evaluating such baselines, the benchmark’s difficulty and relevance remain only partially demonstrated.

**Questions:**

>Each asset is then enlarged with LLM-generated tables designed in enterprise schema style, including realistic names, types, and constraints, while ensuring the question-SQL pairs in the original benchmarks remain valid.

What defines “realistic” names, types, and constraints in this context? What criteria are used to validate the quality of LLM-generated tables? Also, if an LLM is equipped with keyword search, would it still struggle to retrieve relevant content, or could it easily reduce the search scope?


>answer uniqueness, checking whether a question admits only one semantically valid SQL regard- less of syntactic form, as domain-level refinements may introduce semantically similar columns;

How is answer uniqueness verified, especially when schema augmentation introduces similar or redundant columns? Is there an exhaustive check across all tables?

> Unlike BIRD and Spider 2.0-snow/lite, which directly provide a document including a small amount of dense external knowledge as part of the model input.

This seems to understate the difficulty of Spider 2.0, which includes dialect documentation and code. These are not trivial to process and also require retrieval. Could you clarify why these are considered “small” or less challenging compared to your setting?

---

> ### Author Response · Authors · 2025-12-03
>
> We appreciate your recognition of our work and your valuable feedback. Below, we provide our responses to the questions you have raised.
> ## W1) Details of Quality Control
>
> In the revised version of the paper, we will add a detailed description of the quality control pipeline in the appendix. Here we provide a brief overview:
>
> ### Annotation Procedure
>
> We adopt a **double-blind annotation + expert-in-the-loop** strategy to validate three quality indicators. The validation process is conducted by two master’s students with strong SQL expertise and two undergraduate students (student group), as well as one enterprise expert (expert group).
>
> Each sampled instance is randomly assigned to two annotators from the student group for double-blind annotation. If their annotations are inconsistent, the instance is forwarded to the enterprise expert for arbitration and discussion until consensus is reached.
>
> ### Validation of Quality Metrics
>
> 1. **Answer uniqueness**:
>    We provide the full context to the two student annotators, including the user question, the schemas of the top-20 tables ranked by semantic similarity (ensuring coverage of the gold tables), and the external knowledge required to answer the question. The annotators are asked to write down all reasonable SQL answers.
>    If the two annotators produce different numbers of SQL queries or obtain different execution results, the case is escalated to the expert for further judgment.
>    If both annotators provide exactly one SQL query, and the execution result matches the gold SQL result, the answer is marked as unique. All other cases are marked as non-unique.
>
> 2. **Semantic alignment**:
>    This metric verifies whether the SQL query correctly answers the question by checking the execution result and semantic equivalence. This validation task is relatively straightforward for annotators proficient in SQL.
>
> 3. **Document correctness**:
>    This includes checking whether the generated document correctly contains the original external knowledge and whether there exist any contradictions with the source knowledge. This verification is also manageable for annotators who have received formal English education and are proficient with translation tools.
>
> ### Additional Quality Control Measures
>
> Beyond annotation, we introduce many quality-preserving designs during benchmark refinement, including few-shot examples sampled from real enterprises, a complex data synthesis pipeline, and carefully designed prompts. Detailed descriptions can be found in Appendix B and Figures 14–21 of our paper.
>
> ## W2) Clarification of Enterprise Feature Definitions and Benchmark Focus
>
> ### Definition of “Enterprise Features”
>
> As stated in the paper title and the Introduction section, the core enterprise characteristics and challenges distilled in this work include:
>
> - **Massive query scope**: a single question may involve hundreds to tens of thousands of tables.
> - **Complex schemas**: hierarchical and standardized table naming, concrete and heavily abbreviated column naming, and the presence of redundant/legacy tables or columns.
> - **Scattered knowledge**: external knowledge useful for answering the question (e.g., table/column descriptions, value illustrations, domain knowledge) is dispersed across numerous documents.
>
> For the meanings of *“enterprise-style schemas”* and *“enterprise knowledge documents”*, please refer to the refinement details in Appendix B:
>
> - In **Appendix B.1**, when synthesizing schemas, we provide real enterprise schemas as few-shot exemplars to guide the LLM to generate enterprise-style schemas. We then apply schema-level refinement to further ensure that the synthetic schemas follow enterprise patterns.
> - In **Appendix B.3**, we collected common enterprise documentation genres and instructed the LLM to generate extended knowledge following those genres. Typical document types are listed in Table 8.
>
> Regarding the naming transformation rules in schema-level refinement:
> These rules are *not* arbitrarily designed. They are derived from statistical analysis of real enterprise data and from authoritative references.
>
> These conventions have been validated by enterprise experts. Moreover, databases sourced from BigQuery and Snowflake in Spider 2.0 follow similar naming conventions. Such naming practices are extremely common in enterprises (for distinguishing namespaces, enabling cross-project access, supporting multi-tenancy isolation, etc.).
>
> ### Focus of Our Benchmark
>
> It is true that user queries in real enterprise environments can be more complex, but this is *not* the focus of our benchmark. Omitting the complexity of user queries is a trade-off we make to balance benchmark construction cost and coverage.
>
> Nevertheless, our benchmark successfully introduces **three enterprise-specific and unavoidable challenges**, complementing Spider 2.0, BEAVER, and others. Together, these benchmarks form a more complete and diversified Text-to-SQL evaluation ecosystem.

---

> ### Author Response · Authors · 2025-12-03
>
> ## W3) Scope of Evaluation
>
> ### Baselines with Retrieval Capabilities
>
> After carefully reviewing the papers you provided, we note that:
>
> 1. **CHASE-SQL** does not include any explicit table or knowledge retrieval steps.
> 2. **AskData** is not open-sourced, so we cannot access or replicate its schema-linking scoring formula.
> 3. **arXiv:2503.23157** uses schema-linking correctness as a reward signal, but still adopts the traditional full-schema input setting, with *no document collection or large-scale asset setting*. Thus, no retrieval is required, and the method only applies to benchmarks with small query scopes.
> 4. **arXiv:2503.00223** merely uses RL to train a base model (using SQL execution results as rewards). It has no connection to table/knowledge retrieval as defined in our work.
>
> In fact, **current Text-to-SQL methods do not consider knowledge retrieval at all**. Many table-retrieval or schema-linking methods are LLM-driven, but the scale of tables and knowledge in our benchmark far exceeds the context length that current LLM methods can handle. Existing approaches cannot support retrieval at this scale.
>
> ### Keyword Search Tool Baseline
>
> We evaluate BM25 as the keyword search tool and compare keyword-only, embedding-only, and hybrid retrieval. The results are:
>
> 1. Retrieving Top-10 Tables
>
> | Search Method                     | Precision | Recall | F1 Score | Perfect Recall |
> |----------------------------------|-----------|--------|----------|----------------|
> | Embedding-based search tool      | 16.4      | 86.0   | 27.0     | 76.5           |
> | Keyword search tool (BM25)       | 10.5      | 56.4   | 17.4     | 41.7           |
> | Hybrid Search                    | 16.6      | 87.2   | 27.4     | 77.8           |
>
> 2. Retrieving Top-10 Knowledge Documents
>
> | Search Method                | Precision | Recall | F1 Score | Perfect Recall |
> |------------------------------|-----------|--------|----------|----------------|
> | Embedding-based search tool  | 20.8      | 62.9   | 30.2     | 31.8           |
> | Keyword search tool          | 19.3      | 59.3   | 28.1     | 25.5           |
> | Hybrid Search                | 21.2      | 64.2   | 30.8     | 32.7           |
>
> Here, “Embedding-based search tool” denotes using Qwen3-Embedding-0.6B.
> Hybrid search first applies BM25 to retrieve the top-250 tables (96.3 recall) and top-500 knowledge documents (92.9 recall), and then applies embedding retrieval.
>
> The experiments show:
> - Keyword-only retrieval performs significantly worse.
> - Hybrid search improves over embedding-only *only marginally*.
>
> Main reasons:
> 1. After schema-level refinement, field names become highly abbreviated and cannot directly match natural-language tokens.
> 2. Knowledge-level refinement introduces additional enterprise-style narrative that distracts keyword retrieval.
>
> ## Q1) Realism and Quality of LLM-Generated Tables; Keyword-Based Baseline
>
> For realism, see Figure 14:
> We use schemas from real enterprise tables as few-shot examples to guide LLM generation, and schema-level refinement ensures alignment with enterprise practices. Our goal is to **refine academic benchmarks** to better simulate enterprise data environments—not to replicate real enterprise data literally. Although LLM-generated content is synthetic, it more faithfully reflects enterprise challenges than existing benchmarks and is a cost-effective approach to benchmark construction.
>
> For quality, see W1 for details on the quality-control pipeline.
> As described in Appendix B.1, we never execute SQL on the synthetic tables; they only serve as distraction. Therefore, minor logical imperfections are acceptable.
>
> For adding additional baselines, see W3.
>
> ## Q2) Answer Uniqueness Verification
>
> See W1.
>
> ## Q3) Difficulty Comparison with Spider 2.0
>
> Refer to Table 1’s **# Know. Tok. / Question** metric:
>
> - In **Spider 2.0-snow/lite**, each question corresponds to a *single* knowledge document (~340 tokens).
> - In **BIRD-Ent**, each question corresponds to **1,412 documents totaling 1.5M tokens**—far exceeding Spider 2.0 in both length and richness.
>
> Spider 2.0’s knowledge can fit into an LLM’s context window for direct analysis.
> BIRD-Ent’s 1.5M-token knowledge corpus **cannot** be processed directly and requires highly capable retrieval tools. As shown in Section 5.4, even the strongest embedding-retrieval methods struggle in this setting.
>
> Regarding code and dialects in Spider 2.0:
> This represents a different dimension of enterprise difficulty. We do not claim our benchmark is universally more challenging than Spider 2.0, but that it is **more challenging along the enterprise characteristics we identify**. Our goal is to complement existing benchmarks—not to replace them or pursue extreme difficulty.

---

### Official Review · Reviewer_Nq2W · 2025-10-30

**Soundness:** 2
**Presentation:** 3
**Contribution:** 2
**Rating:** 4
**Confidence:** 3

**Summary:**

The paper introduces enterprise-oriented versions of Text-to-SQL datasets (BIRD-Ent and Spider-Ent) to better reflect Text-to-SQL retrieval and generation challenges under massive schemas, tables, and knowledge documents. The dataset construction approach consists of three stages: (i) domain-level refinement, where schemas and tables from similar domains are grouped together, (ii) schema-level refinement, where schema naming conventions are modified to follow enterprise standards, and (iii) knowledge-level refinement, where an LLM augments knowledge sources by generating related documents. Experimental results across different LLMs demonstrate that BIRD-Ent and Spider-Ent datasets are more challenging than the original datasets under different LLMs.

**Strengths:**

- S1) Experimental results demonstrate that Ent-versions of BIRD and Spider datasets present greater challenges across multiple LLMs (Table 2). Notably, LLMs continue to underperform even when provided with oracle inputs (Table 5).
- S2) LLM-based schema-level refinement enables hierarchical decomposition of schemas into structured components (<project> <area> <content>), facilitating systematic merging to construct larger, more complex schemas.
- S3) The paper implements quality control mechanisms that leverage manual human annotation to assess benchmark quality and ensure question-answer correctness is maintained.

**Weaknesses:**

- W1) The paper claims that scaled-up datasets reflect enterprise realities, but this assertion lacks external validation from SQL experts or enterprise practitioners. For example, Table 1's claim that Spider-Ent demonstrates superior "Enterprise Realism" compared to BEAVER is not systematically verified: Appendix A.1 provides a supporting example, but this represents anecdotal evidence rather than systematic real-world verification.
- W2) The dataset construction generates synthetic tables to expand database size, but relies heavily on LLM parametric knowledge for table creation (Figure 14). This approach raises concerns about applicability to realistic scenarios, such as generating tables with customer transaction data that are not part of LLMs' training data. As a result, it is unclear whether the generated tables actually enhance benchmark quality or simply introduce noise, artificially increasing complexity.
- W3) Knowledge-level refinement using DeepSeek-generated documents appears to function as an adversarial approach to degrade retrieval performance by creating numerous documents with highly similar embeddings. The authors should present an analysis examining how embedding similarity distributions shift with their synthetic document generation and  and whether this creates retrieval noise.

**Questions:**

- Q1) Could the LLMs fail in the oracle setting due to the long input of the generated schemas/tables and the redundant noise introduced by the synthetic database expansion?

- Q2) How do you ensure that the synthetically generated tables and documents do not simply introduce noise to the dataset? Could we use similar synthetic data generation to improve the database quality instead of introducing noise to it?

---

> ### Author Response · Authors · 2025-12-03
>
> We sincerely appreciate your recognition of our work and your valuable feedback. Below, we provide our responses and clarifications to the weaknesses and questions you have raised.
> ## W1) Systematic Validation of Enterprise Realism
> We ensure the *enterprise realism* of our benchmark through multiple perspectives, including analyses and validations grounded in real-world enterprise data.
>
> 1. **Statistical analysis over 324 enterprise data asset domains and multiple benchmarks.**
>
>    - We sampled 2,000 tables from real enterprise data assets to analyze enterprise-specific naming characteristics, including long names, hierarchical naming, extensive abbreviations, and domain-specific terminology:
>
>      |                           | Real Enterprise | Spider-Ent | BIRD-Ent | BEAVER | Spider 2.0-snow/lite | Spider 2.0-DBT | Spider | BIRD |
>      |---------------------------|------------------|------------|----------|--------|-----------------------|----------------|--------|------|
>      | Avg. Table name length (tokens) | **10.2** | 5.2 | 5.2 | 2.6 | 8.3 | 3.3 | 1.4 | 1.2 |
>      | Hierarchical naming       | **Y**            | Y | Y | N | Y | N | N | N |
>      | Avg. Column name length (tokens) | **3.5** | 2.1 | 2.1 | 2.1 | 3.75 | 2.3 | 1.8 | 1.9 |
>      | Avg. token Zipf freq.     | **3.92** | 4.25 | 4.25 | 4.55 | 4.71 | 4.55 | 4.91 | 4.66 |
>
>      `Avg. token Zipf freq.` is computed by splitting all table and column names into tokens, obtaining the Zipf frequency of each token using the *wordfreq* library (higher Zipf values indicate more common English words), and then averaging these values across the dataset. Both real enterprise data and our Ent-series benchmarks show relatively low values on this metric, because real enterprise schemas contain a large number of abbreviations and domain-specific terms.
>
>    - We also analyzed the query scope of real enterprises:
>
>      |                     | Real Enterprise | BIRD-Ent | Spider 2.0-snow/lite | Others |
>      |---------------------|------------------|----------|----------------------|--------|
>      | # Table / QS        | 625.8            | 392.1    | 50.4                 | See Table 1 in the paper for more details. |
>
> 2. **Expert consultation and blind evaluation.**
>
>    During the preliminary investigation phase before formally constructing the benchmark, we conducted in-depth discussions with multiple enterprise experts and collected a large number of real-world use cases. Our characterization of enterprise scenarios was recognized by these experts and validated against concrete examples.
>
>    After the benchmarks were constructed, we further invited 4 enterprise experts to conduct blind evaluations on multiple benchmarks. The evaluation considers seven dimensions: task setting, schema scale, schema style, knowledge scale, knowledge content, natural language questions, and SQL queries. The benchmarks were ranked according to their closeness to real enterprise scenarios. The final results are shown below:
>
>    | Metric | Spider-Ent | BIRD-Ent | BEAVER | Spider 2.0-lite/snow | Spider 2.0-DBT | Spider | BIRD |
>    |--------|------------|----------|--------|-----------------------|----------------|--------|------|
>    | Average Rank Score | 3.1 | 1.6 | 3.4 | 2.0 | 2.6 | 4.3 | 3.6 |
>    | Final Rank | 4 | 1 | 5 | 2 | 3 | 7 | 6 |
>
> 3. **Support from authoritative references.**
>
>    - Recent work (DOI: 10.18653/v1/2025.emnlp-main.51, arXiv: 2507.23104) discusses the large-scale schema issue commonly observed in real-world and enterprise databases.
>    - Official documents from Oracle and Alibaba Cloud
>      (https://docs.oracle.com/cd/E38317_01/doc.11117/e37988/namingconv.htm,
>      https://www.alibabacloud.com/help/en/dataworks/user-guide/data-warehouse-layering)
>      recommend or directly adopt hierarchical table naming conventions similar to those used in our benchmark.
>    - Documentation from SAP BW Consulting
>      (https://www.sapbwconsulting.com/blog/data-warehouse-naming-conventions-best-practices-for-sap)
>      indicates that enterprise data warehouses follow column and table naming principles similar to ours, including richer business semantics and finer granularity.
>    - A recent study (DOI: 10.18653/v1/2025.findings-emnlp.1348) reports extensive use of abbreviations in real enterprise table and column names, consistent with our benchmark design.
>    - Another work (DOI: 10.18653/v1/2025.acl-demo.27) highlights the existence of legacy tables caused by version evolution in real-world databases, corresponding to the redundant tables in our benchmark.
>
> 4. **Industry consensus.**
>
>    Massive query scope, complex database schemas, and knowledge scattered across documents are natural data characteristics of large-scale enterprise data systems, arising from cross-system integration, data life cycles, and non-standardized processes.

---

> ### Author Response · Authors · 2025-12-03
>
> ## W2) Do the Synthetic Tables Improve Benchmark Quality or Introduce Noise?
>
> First, based on the systematic analysis and validation in W1, the characteristics we intentionally introduce are consistent with our claim of being *enterprise-level*, rather than merely injecting noise.
>
> Second, from the objective of our benchmark construction, the purpose of synthesizing tables and knowledge documents is to introduce **structural challenges**, rather than challenges rooted in specific data values or domain semantics. Our benchmark focuses on whether models can handle **massive query scope**, **complex schemas**, and **scattered knowledge**, which are core characteristics of enterprise environments (and we have demonstrated that these goals are achieved). It does not aim to evaluate the understanding of business logic such as that in “customer transaction tables.” Therefore, the synthetic tables do not need to preserve *industry-realistic content*, but only need to preserve *enterprise-style structural characteristics*.
>
> To this end, we use real enterprise schemas as few-shot exemplars during table synthesis to ensure that the structures generated by LLMs are consistent with enterprise patterns (see Figure 14), thereby guaranteeing benchmark quality.
>
> In addition, Section 3 of our paper describes the quality verification process in detail. In Appendix B, which presents the refinement pipeline, we further introduce multiple safeguards to ensure that the newly added content is non-destructive to the benchmark.
>
>
> ---
>
> ## W3) Embedding Similarity Distribution of the Synthetic Knowledge Documents
>
> First, we clarify that our **knowledge-level refinement** does not deliberately generate a large number of highly similar documents in the embedding space. Instead, it expands the original external knowledge and database descriptions in BIRD (which are typically short) into long-form documents that resemble common enterprise documentation genres. Our goal is to *sparsify* the original knowledge through textual expansion, rather than to perform adversarial data augmentation.
>
> To fully address your concern, we treat the original BIRD knowledge as *initial documents* and conduct an embedding similarity analysis between the original documents and their refined counterparts (using Qwen3-Embedding-0.6B with cosine similarity):
>
> ### Analysis 1: Similarity Between Each Original Document and Its Refined Counterpart
>
> We compute the similarity between each original document and its refined counterpart. The results show that the average similarity is 0.65 with a standard deviation of 0.08, indicating a moderate level of semantic shift. This suggests that the refined documents inherit the core semantics of the original documents while introducing additional enterprise-style narrative content.
>
> ### Analysis 2: Pairwise Embedding Similarity Distributions Within Original and Refined Documents
>
> We further compute pairwise similarities within each document group to examine whether refinement introduces abnormal effects (e.g., excessive duplication or reduced diversity):
>
> |                    | Average | Std  | P(sim > 0.8) |
> |--------------------|---------|------|--------------|
> | Original documents | 0.46    | 0.07 | 0.0030       |
> | Refined documents  | 0.32    | 0.09 | 0.0001       |
>
> The results indicate that:
> - Inter-document embedding similarity decreases after refinement. Rather than making documents more redundant, refinement increases their semantic diversity. The average similarity among refined documents is only 0.32.
> - The probability of P(sim > 0.8) for refined documents is almost zero, indicating that no large clusters of highly similar documents (i.e., abnormal repetition) emerge.
> - The standard deviation remains stable, showing that refinement does not harm knowledge diversity.
>
> Therefore, knowledge-level refinement is **not noise injection**, but a form of **textual reconstruction**, designed to increase document length and dilute key information.
>
> In addition, we analyzed 20 knowledge-retrieval failure cases and summarized the main failure patterns:
> 1. The additional enterprise-style narrative reduces the overall semantic similarity between the document and the question, even though some local segments (originating from the original knowledge) still have high similarity.
> 2. The semantic association between the question and the knowledge is highly implicit. For example, the question concerns disease diagnosis (implicitly referencing biochemical indicators), while the interpretation rules of those indicators are stored in the knowledge documents.
> 3. After schema-level refinement, column names become heavily abbreviated, making it difficult for natural language entities in the question to align with schema elements.

---

> ### Author Response · Authors · 2025-12-03
>
> ## Q1) Could the LLMs Fail in the Oracle Setting Due to Long Generated Schemas and Redundant Noise?
>
> Yes. Even in the oracle setting, LLMs frequently fail. Please refer to Table 5 for the oracle-setting results and Figure 11 for concrete failure cases.
>
>
> ---
>
> ## Q2.1) How Do You Ensure That the Synthetically Generated Tables and Documents Do Not Simply Introduce Noise?
>
> See W2.
>
>
> ---
>
> ## Q2.2) Could Similar Synthetic Data Generation Be Used to Improve Database Quality Instead of Introducing Noise?
>
> Yes. OmniSQL (https://doi.org/10.14778/3749646.3749723, VLDB’25) adopts a similar LLM-based synthesis framework to generate large-scale Text-to-SQL data, including databases. Their analysis shows that the synthetically generated databases even surpass the human-constructed BIRD databases in terms of quality, and models fine-tuned on the synthesized data achieve substantial performance improvements.

---

### Official Review · Reviewer_9mMo · 2025-11-03

**Soundness:** 2
**Presentation:** 3
**Contribution:** 2
**Rating:** 4
**Confidence:** 4

**Summary:**

The paper proposes a new nl2sql benchmark by extending bird and spider benchmarks. It does so by synthetically augmenting/modifying the data to imitate more complex interprise setting. The paper shows the constructed datasets have many more tables/need for external knowledge, and that the accuracy of existing methods drop when run on the refined benchmark compared with the original benchmark data.

**Strengths:**

- New benchmarks are always good, especially if they mimic real-world settings

- The method for refining the data using LLMs is interesting

**Weaknesses:**

- There are two major issues I have with the paper. First, it is unclear how realistic the benchmark is. The refinement steps are motivated by "In enterprise settings, .....". Where is this information coming from about enterprise setting? Why is it representative of all enterprise use-cases? The main metrics shown at the end are number of table/columns/knowledge tokens. However, the data distribution/content is important and not only the size.

- Second, the data quality is unclear. There is significant issues with existing text-2-sql benchmark ground-truth values, and the paper needs to do its due diligence to ensure the ground-truth answers are correct. Right now, the paper is very handwavy about the process of obtaining ground-truth values. They inspected 10% of the samples, which isn't enough, and even for it is unclear how. The paper mentions "with double annotation and expert arbitration". Who are the annotators? who are the experts? What was the annotation process?

**Questions:**

Please respond to the two points raise above.

---

> ### Author Response · Authors · 2025-12-03
>
> ## W1.1) Information Sources for Enterprise Scenarios
>
> Our information primarily comes from the data intelligence platform department of a large technology company, which has accumulated a substantial amount of enterprise-level database resources and real-world Text-to-SQL use cases through its business practices. In addition, we also refer to authoritative literature and widely accepted industry consensus.
>
> We ensure the *enterprise realism* of our benchmark from the following aspects:
>
> 1. **Statistical analysis over 324 enterprise data asset domains and multiple benchmarks.**
>
>    - We sampled 2,000 tables from real enterprise data assets to analyze enterprise-specific naming characteristics, including long names, hierarchical naming, extensive abbreviations, and domain-specific terminology:
>
>      |                           | Real Enterprise | Spider-Ent | BIRD-Ent | BEAVER | Spider 2.0-snow/lite | Spider 2.0-DBT | Spider | BIRD |
>      |---------------------------|------------------|------------|----------|--------|-----------------------|----------------|--------|------|
>      | Avg. Table name length (tokens) | **10.2** | 5.2 | 5.2 | 2.6 | 8.3 | 3.3 | 1.4 | 1.2 |
>      | Hierarchical naming       | **Y**            | Y | Y | N | Y | N | N | N |
>      | Avg. Column name length (tokens) | **3.5** | 2.1 | 2.1 | 2.1 | 3.75 | 2.3 | 1.8 | 1.9 |
>      | Avg. token Zipf freq.     | **3.92** | 4.25 | 4.25 | 4.55 | 4.71 | 4.55 | 4.91 | 4.66 |
>
>      `Avg. token Zipf freq.` is computed by splitting all table and column names into tokens, obtaining the Zipf frequency of each token using the *wordfreq* library (higher Zipf values indicate more common English words), and then averaging these values across the dataset. Both real enterprise data and our Ent-series benchmarks show relatively low values on this metric, because real enterprise schemas contain a large number of abbreviations and domain-specific terms.
>
>    - We also analyzed the query scope of real enterprises:
>
>      |                     | Real Enterprise | BIRD-Ent | Spider 2.0-snow/lite | Others |
>      |---------------------|------------------|----------|----------------------|--------|
>      | # Table / QS        | 625.8            | 392.1    | 50.4                 | See Table 1 in the paper for more details. |
>
> 2. **Expert consultation and blind evaluation.**
>
>    During the preliminary investigation phase before formally constructing the benchmark, we conducted in-depth discussions with multiple enterprise experts and collected a large number of real-world use cases. Our characterization of enterprise scenarios was recognized by these experts and validated against concrete examples.
>
>    After the benchmarks were constructed, we further invited 4 enterprise experts to conduct blind evaluations on multiple benchmarks. The evaluation considers seven dimensions: task setting, schema scale, schema style, knowledge scale, knowledge content, natural language questions, and SQL queries. The benchmarks were ranked according to their closeness to real enterprise scenarios. The final results are shown below:
>
>    | Metric | Spider-Ent | BIRD-Ent | BEAVER | Spider 2.0-lite/snow | Spider 2.0-DBT | Spider | BIRD |
>    |--------|------------|----------|--------|-----------------------|----------------|--------|------|
>    | Average Rank Score | 3.1 | 1.6 | 3.4 | 2.0 | 2.6 | 4.3 | 3.6 |
>    | Final Rank | 4 | 1 | 5 | 2 | 3 | 7 | 6 |
>
> 3. **Support from authoritative references.**
>
>    - Recent work (DOI: 10.18653/v1/2025.emnlp-main.51, arXiv: 2507.23104) discusses the large-scale schema issue commonly observed in real-world and enterprise databases.
>    - Official documents from Oracle and Alibaba Cloud
>      (https://docs.oracle.com/cd/E38317_01/doc.11117/e37988/namingconv.htm,
>      https://www.alibabacloud.com/help/en/dataworks/user-guide/data-warehouse-layering)
>      recommend or directly adopt hierarchical table naming conventions similar to those used in our benchmark.
>    - Documentation from SAP BW Consulting
>      (https://www.sapbwconsulting.com/blog/data-warehouse-naming-conventions-best-practices-for-sap)
>      indicates that enterprise data warehouses follow column and table naming principles similar to ours, including richer business semantics and finer granularity.
>    - A recent study (DOI: 10.18653/v1/2025.findings-emnlp.1348) reports extensive use of abbreviations in real enterprise table and column names, consistent with our benchmark design.
>    - Another work (DOI: 10.18653/v1/2025.acl-demo.27) highlights the existence of legacy tables caused by version evolution in real-world databases, corresponding to the redundant tables in our benchmark.
>
> 4. **Industry consensus.**
>
>    Massive query scope, complex database schemas, and knowledge scattered across documents are natural data characteristics of large-scale enterprise data systems, arising from cross-system integration, data life cycles, and non-standardized processes.

---

> ### Author Response · Authors · 2025-12-03
>
> ## W1.2) Representativeness of Enterprise Scenarios
>
> Real enterprise data are difficult to release publicly due to regulations, privacy protection, and confidentiality requirements. No single enterprise dataset can represent all enterprise scenarios. Both BEAVER and Spider 2.0 are constructed from data and use cases of only a few specific enterprises.
>
> Given this inherent diversity, the meaningful goal of enterprise-oriented benchmarks is not to replicate every enterprise environment, but to identify and model the key challenge dimensions that repeatedly occur in large-scale real-world systems. In our work, these dimensions are *massive scope*, *complex schema*, and *scattered knowledge*. Our benchmark does not aim to evaluate the understanding of enterprise-specific business logic, and therefore does not require matching the underlying data distributions or business semantics.
>
> Our contribution lies in identifying and amplifying concrete enterprise challenges that are well documented in industry practice (see W1.1), yet have not been sufficiently reflected in existing benchmarks. Enterprise-level Text-to-SQL ultimately requires a diversified benchmark ecosystem. Our goal is to complement this ecosystem rather than to replace any existing benchmark entirely.
>
> ---
>
> ## W2) Details of Quality Control
>
> In the revised version of the paper, we will add a detailed description of the quality control pipeline in the appendix. Here we provide a brief overview:
>
> ### Annotation Procedure
>
> We adopt a **double-blind annotation + expert-in-the-loop** strategy to validate three quality indicators. The validation process is conducted by two master’s students with strong SQL expertise and two undergraduate students (student group), as well as one enterprise expert (expert group).
>
> Each sampled instance is randomly assigned to two annotators from the student group for double-blind annotation. If their annotations are inconsistent, the instance is forwarded to the enterprise expert for arbitration and discussion until consensus is reached.
>
> ### Validation of Quality Metrics
>
> 1. **Answer uniqueness**:
>    We provide the full context to the two student annotators, including the user question, the schemas of the top-20 tables ranked by semantic similarity (ensuring coverage of the gold tables), and the external knowledge required to answer the question. The annotators are asked to write down all reasonable SQL answers.
>    If the two annotators produce different numbers of SQL queries or obtain different execution results, the case is escalated to the expert for further judgment.
>    If both annotators provide exactly one SQL query, and the execution result matches the gold SQL result, the answer is marked as unique. All other cases are marked as non-unique.
>
> 2. **Semantic alignment**:
>    This metric verifies whether the SQL query correctly answers the question by checking the execution result and semantic equivalence. This validation task is relatively straightforward for annotators proficient in SQL.
>
> 3. **Document correctness**:
>    This includes checking whether the generated document correctly contains the original external knowledge and whether there exist any contradictions with the source knowledge. This verification is also manageable for annotators who have received formal English education and are proficient with translation tools.
>
> ### Additional Quality Control Measures
>
> Beyond annotation, we introduce many quality-preserving designs during benchmark refinement, including few-shot examples sampled from real enterprises, a complex data synthesis pipeline, and carefully designed prompts. Detailed descriptions can be found in Appendix B and Figures 14–21 of our paper.
>
> ### Ground-truth Verification
>
> Our benchmark refinement is based on two of the most influential Text-to-SQL benchmarks: **BIRD** and **Spider**, both of which have been widely adopted by existing methods (e.g., Din-SQL, XiYan-SQL) and have received substantial community validation. Their questions and ground-truth SQL queries are entirely human-annotated and have undergone multiple rounds of verification.
>
> Our refinement process does not modify the original question–answer pairs (table and column renaming can be mapped one-to-one). Instead, we only increase task difficulty by restructuring the data environment. Therefore, the probability of affecting the correctness of the ground truth is very low (see Section 3 on quality control).

---

### Author Response · Authors · 2025-12-03
**Discussion Summary 1**

Below we summarize the reviewers’ most common concerns and provide brief responses (detailed questions and responses can be found in the replies to individual reviewers), to help the AC/PC and other interested readers quickly grasp the core points of the discussion.

---

## Q1. How Do We Systematically Ensure That Our Benchmark Reflects Real Enterprise Scenarios?

We ensure the *enterprise realism* of our benchmark from multiple perspectives, including analyses and validations grounded in real-world evidence：

1. **Statistical analysis over 324 enterprise data asset domains and multiple benchmarks.**

   - We sampled 2,000 tables from real enterprise data assets to analyze enterprise-specific naming characteristics, including long names, hierarchical naming, extensive abbreviations, and domain-specific terminology:

     |                           | Real Enterprise | Spider-Ent | BIRD-Ent | BEAVER | Spider 2.0-snow/lite | Spider 2.0-DBT | Spider | BIRD |
     |---------------------------|------------------|------------|----------|--------|-----------------------|----------------|--------|------|
     | Avg. Table name length (tokens) | **10.2** | 5.2 | 5.2 | 2.6 | 8.3 | 3.3 | 1.4 | 1.2 |
     | Hierarchical naming       | **Y**            | Y | Y | N | Y | N | N | N |
     | Avg. Column name length (tokens) | **3.5** | 2.1 | 2.1 | 2.1 | 3.75 | 2.3 | 1.8 | 1.9 |
     | Avg. token Zipf freq.     | **3.92** | 4.25 | 4.25 | 4.55 | 4.71 | 4.55 | 4.91 | 4.66 |

     `Avg. token Zipf freq.` is computed by splitting all table and column names into tokens, obtaining the Zipf frequency of each token using the *wordfreq* library (higher Zipf values indicate more common English words), and then averaging these values across the dataset. Both real enterprise data and our Ent-series benchmarks show relatively low values on this metric, because real enterprise schemas contain a large number of abbreviations and domain-specific terms.

   - We also analyzed the query scope of real enterprises:

     |                     | Real Enterprise | BIRD-Ent | Spider 2.0-snow/lite | Others |
     |---------------------|------------------|----------|----------------------|--------|
     | # Table / QS        | 625.8            | 392.1    | 50.4                 | See Table 1 in the paper for more details. |

2. **Expert consultation and blind evaluation.**

   During the preliminary investigation phase before formally constructing the benchmark, we conducted in-depth discussions with multiple enterprise experts and collected a large number of real-world use cases. Our characterization of enterprise scenarios was recognized by these experts and validated against concrete examples.

   After the benchmarks were constructed, we further invited 4 enterprise experts to conduct blind evaluations on multiple benchmarks. The evaluation considers seven dimensions: task setting, schema scale, schema style, knowledge scale, knowledge content, natural language questions, and SQL queries. The benchmarks were ranked according to their closeness to real enterprise scenarios. The final results are shown below:

   | Metric | Spider-Ent | BIRD-Ent | BEAVER | Spider 2.0-lite/snow | Spider 2.0-DBT | Spider | BIRD |
   |--------|------------|----------|--------|-----------------------|----------------|--------|------|
   | Average Rank Score | 3.1 | 1.6 | 3.4 | 2.0 | 2.6 | 4.3 | 3.6 |
   | Final Rank | 4 | 1 | 5 | 2 | 3 | 7 | 6 |

3. **Support from authoritative references.**

   - Recent work (DOI: 10.18653/v1/2025.emnlp-main.51, arXiv: 2507.23104) discusses the large-scale schema issue commonly observed in real-world and enterprise databases.
   - Official documents from Oracle and Alibaba Cloud
     (https://docs.oracle.com/cd/E38317_01/doc.11117/e37988/namingconv.htm,
     https://www.alibabacloud.com/help/en/dataworks/user-guide/data-warehouse-layering)
     recommend or directly adopt hierarchical table naming conventions similar to those used in our benchmark.
   - Documentation from SAP BW Consulting
     (https://www.sapbwconsulting.com/blog/data-warehouse-naming-conventions-best-practices-for-sap)
     indicates that enterprise data warehouses follow column and table naming principles similar to ours, including richer business semantics and finer granularity.
   - A recent study (DOI: 10.18653/v1/2025.findings-emnlp.1348) reports extensive use of abbreviations in real enterprise table and column names, consistent with our benchmark design.
   - Another work (DOI: 10.18653/v1/2025.acl-demo.27) highlights the existence of legacy tables caused by version evolution in real-world databases, corresponding to the redundant tables in our benchmark.

4. **Industry consensus.**

   Massive query scope, complex database schemas, and knowledge scattered across documents are natural data characteristics of large-scale enterprise data systems, arising from cross-system integration, data life cycles, and non-standardized processes.

---

> ### Author Response · Authors · 2025-12-03
> **Discussion Summary 2**
>
> ## Q2. What Are the Details of Our Quality Control Pipeline?
> ### Annotation Procedure
>
> We adopt a **double-blind annotation + expert-in-the-loop** strategy to validate three quality indicators. The validation process is conducted by two master’s students with strong SQL expertise and two undergraduate students (student group), as well as one enterprise expert (expert group).
>
> Each sampled instance is randomly assigned to two annotators from the student group for double-blind annotation. If their annotations are inconsistent, the instance is forwarded to the enterprise expert for arbitration and discussion until consensus is reached.
>
> ### Validation of Quality Metrics
>
> 1. **Answer uniqueness**:
>    We provide the full context to the two student annotators, including the user question, the schemas of the top-20 tables ranked by semantic similarity (ensuring coverage of the gold tables), and the external knowledge required to answer the question. The annotators are asked to write down as many reasonable SQL answers as possible.
>    If the two annotators produce a different number of SQL queries or obtain different execution results, the case is escalated to the expert for arbitration.
>    If both annotators provide exactly one SQL query, and the execution result matches the gold SQL result, the answer is marked as unique. All other cases are marked as non-unique.
>
> 2. **Semantic alignment**:
>    This metric verifies whether the SQL query correctly answers the question by checking the execution result and semantic equivalence. This validation is relatively straightforward for annotators proficient in SQL.
>
> 3. **Document correctness**:
>    This includes checking whether the generated document correctly contains the original external knowledge and whether there exist any contradictions with the source knowledge. This verification is also manageable for annotators who have received formal English education and are proficient with translation tools.
>
> ### Additional Quality Control Designs
>
> During benchmark refinement, we introduce many quality-preserving designs (e.g., few-shot exemplars sampled from real enterprises, a complex data synthesis pipeline, and carefully designed prompts). Readers may refer to **Appendix B** and **Figures 14–21** for detailed refinement procedures.
>
> In addition, some reviewers (HwqY, 9mMo) noted that a 10% sampling rate for manual inspection may be insufficient. We will therefore increase the sampling size in future updates to further strengthen quality assurance.

---

> ### Author Response · Authors · 2025-12-03
> **Discussion Summary 3**
>
> ## Common Reviewer Misunderstandings
>
> Below we list several typical misunderstandings raised by reviewers and provide our clarifications:
>
> 1. **“The experiments lack SOTA Text-to-SQL systems with integrated retrieval as baselines.”**
>    Reviewer HwqY listed a series of works claimed to be “Text-to-SQL methods with integrated retrieval.” However, after careful inspection, we found that these works either do not explicitly involve table or knowledge retrieval components, or lack reproducibility.
>    In fact, to the best of our knowledge, **existing Text-to-SQL methods generally do not consider knowledge retrieval at all**. Most works on table retrieval or schema-linking are LLM-driven, whereas the scale of table and knowledge contexts in our benchmark far exceeds current LLM context windows. Existing methods are therefore fundamentally incapable of supporting retrieval at this scale.
>
> 2. **“The benchmark design and comparison do not consider data distributions and content.”**
>    We clarify that, in addition to query scope and knowledge scale, we do take data distribution and content into account to a certain extent through multiple measures, including:
>    - Mimicking enterprise-style table and column naming conventions and introducing redundant schemas,
>    - Using real enterprise schemas as few-shot exemplars when synthesizing tables,
>    - Constructing documents using common enterprise writing genres.
>
>    Moreover, from the objective of our benchmark construction, we synthesize tables and knowledge documents to introduce **structural challenges**, rather than challenges rooted in specific data values or domain semantics. Our benchmark focuses on whether models can handle **massive query scope**, **complex schema**, and **scattered external knowledge**, which we have already achieved. It does not aim to evaluate the understanding of business logic such as that in “customer transaction tables.” Therefore, our synthetic content does not need to strictly preserve *enterprise-realistic content*, but rather needs to preserve *enterprise-style structural characteristics*.
>
> 3. **“Our benchmark is harder than Spider 2.0 and BEAVER and can replace them.”**
>    Some reviewers interpreted our motivation as constructing a benchmark that is more difficult than existing enterprise benchmarks and replacing them. We clarify that **we never claim our benchmark is universally more difficult**, and in fact, its focus on enterprise characteristics is not identical to that of Spider 2.0 or BEAVER.
>    Our contribution lies in **identifying and amplifying specific enterprise challenges** that are well documented in industry practice (see Q1), yet are not sufficiently reflected in existing benchmarks. Enterprise Text-to-SQL ultimately requires a **diversified benchmark ecosystem**. Our goal is to **complement this ecosystem**, rather than to replace any single existing benchmark.

---

### Note · Authors · 2026-01-05

I have read and agree with the venue's withdrawal policy on behalf of myself and my co-authors.